# Variational Inference via Rényi Bound Optimization and Multiple-Source Adaptation [note 1]

**DOI:** 10.3390/e25101468

**Published:** 2023-10-20

**Authors:** Dana Zalman (Oshri), Shai Fine

**Affiliations:** 1School of Computer Science, Reichman University, Herzliya 4610101, Israel; 2Data Science Institute, Reichman University, Herzliya 4610101, Israel; shai.fine@runi.ac.il

**Keywords:** multiple-source adaptation, variational inference, Rényi divergence

## Abstract

Variational inference provides a way to approximate probability densities through optimization. It does so by optimizing an upper or a lower bound of the likelihood of the observed data (the evidence). The classic variational inference approach suggests maximizing the Evidence Lower Bound (ELBO). Recent studies proposed to optimize the variational Rényi bound (VR) and the χ upper bound. However, these estimates, which are based on the Monte Carlo (MC) approximation, either underestimate the bound or exhibit a high variance. In this work, we introduce a new upper bound, termed the Variational Rényi Log Upper bound (VRLU), which is based on the existing VR bound. In contrast to the existing VR bound, the MC approximation of the VRLU bound maintains the upper bound property. Furthermore, we devise a (sandwiched) upper–lower bound variational inference method, termed the Variational Rényi Sandwich (VRS), to jointly optimize the upper and lower bounds. We present a set of experiments, designed to evaluate the new VRLU bound and to compare the VRS method with the classic Variational Autoencoder (VAE) and the VR methods. Next, we apply the VRS approximation to the Multiple-Source Adaptation problem (MSA). MSA is a real-world scenario where data are collected from multiple sources that differ from one another by their probability distribution over the input space. The main aim is to combine fairly accurate predictive models from these sources and create an accurate model for new, mixed target domains. However, many domain adaptation methods assume prior knowledge of the data distribution in the source domains. In this work, we apply the suggested VRS density estimate to the Multiple-Source Adaptation problem (MSA) and show, both theoretically and empirically, that it provides tighter error bounds and improved performance, compared to leading MSA methods.

## 1. Introduction

In numerous practical situations, we encounter probability distributions that are challenging to calculate. This occurs especially when the distribution includes hidden variables. Therefore, it becomes necessary to employ approaches that can estimate or approximate such distributions. Variational inference (VI) is a technique used to accomplish this task. VI is a compelling approach for approximating posterior distributions in latent variable models [1]. It can handle intractable and possibly high-dimensional posteriors, and it makes Bayesian inference computationally efficient and scalable to large datasets. To this end, VI defines a simple distribution family, called the variational family, and then finds the optimal member of the variational family that is closest to the true posterior distribution. This transforms the posterior inference into an optimization problem concerning the variational distribution.

One of the most successful applications of VI in the deep neural network realm is the Variational Autoencoder (VAE) [2], which is a deep generative model that implements a probabilistic model and variational Bayesian inference. Many techniques have been suggested to improve the accuracy and efficiency of variational methods (cf. [3,4,5,6,7]). Recent trends in variational inference have focused on the following aspects:Scalability: includes stochastic approximations.Generalization: extends the applicability of VI to a large class of otherwise intractable models, such as non-conjugate models.Accuracy: includes variational models beyond the mean field approximation.Amortization: implements the inference over local latent variables with inference networks.Robustness: generating a reliable representation of particular data types in the encoded space when using corrupted training data and detecting anomalies.

There are other methods for improving approximation such as Monte Carlo methods for VI and black-box methods [8].

In this work, we focus on the accuracy of the VAE models. An essential aspect of the VI methodology revolves around selecting an appropriate divergence method. This divergence measure allows us to approximate the true posterior distribution with a simpler variational distribution. Consequently, the selection of the divergence measure can have a notable impact on the accuracy of the approximation. Furthermore, using the selected divergence measure, one can devise lower and upper bounds, and estimate the true posterior.

Accordingly, we propose a new upper bound for the evidence, termed the Variational Rényi Log Upper bound (VRLU), based on the Variational Rényi (VR) bound suggested by Li and Turner [3]. Further, we devise a (sandwiched) upper–lower bound variational inference method, termed VRS, to jointly optimize the Rényi upper and lower bounds. The VRS loss function combines the VR lower bound and our new upper bound, thus providing a tighter estimate for the log evidence.

Next, we will demonstrate the practical effectiveness of VRS by applying it to the domain adaptation problem. Through this application, we aim to showcase the tangible benefits and practical relevance of our approach.

Domain adaptation is a scenario that arises when we aim to learn from a source data distribution; a well-performing model on a different (but related) target data distribution. A real-world example of domain adaptation is the common spam filtering problem. This problem consists of adapting a model from one user (the source distribution) to a new user who receives significantly different emails (the target distribution).

In the context of domain adaptation, the terms “source” and “target” domains are used to refer to the training and test sets, respectively. These sets can have distinct feature spaces, which can occur when the statistical properties of a domain change over time or when new samples are collected from various sources, resulting in domain shifts. Multiple-Source Adaptation (MSA) addresses scenarios where there are multiple source domains and one target domain. The central question is whether the learner can effectively combine relatively accurate predictors from each source domain to create an accurate predictor for any new target domain that may consist of a mixture of these sources.

In contrast to the majority of machine learning research, where models are trained and tested on data drawn from the same distribution, domain adaptation involves using data from different distributions for training and testing. When the train and test sets share the same distribution, the uniform convergence theory ensures that a model’s empirical training error closely approximates its true error. This assumption is not guaranteed in the MSA problem.

In this work, we have focused on two main ideas:Improving the estimation of the domain distribution using VAE.Using the improved estimated distributions in the algorithm presented in [9] to solve the MSA problem.

The rest of the paper is organized as follows: Section 2 provides a review of variational inference for probabilistic modeling, and discusses different divergence methods such as KL Divergence, Rényi Divergence, and χ Divergence, for bounding the log evidence. In Section 3, we present our novel approach, called Variational Rényi Log Upper bound (VRLU), which offers an improved bound for the log evidence. Additionally, we introduce an optimized technique, referred to as the Variational Rényi Sandwich (VRS), that leverages both upper and lower bounds. Section 4 offers a comprehensive overview of the domain adaptation problem and illustrates the application of the approximated distributions in calculating its loss function. Finally, in Section 5, we present a series of experiments conducted to evaluate the effectiveness of our proposed methods, VRLU and VRS, in the context of both log evidence estimation and domain adaptation.

## 2. Divergence Methods in Variational Inference for Probabilistic Modeling


In probabilistic modeling, we aim to devise a probabilistic model, $p_{\theta }$, that best explains the data. This is commonly done by maximizing the log-likelihood of the data (also known as *log evidence*), with respect to the model’s parameter θ, i.e., Maximum Likelihood Estimation (MLE). For a latent model, where we assume that the observed data, *x*, depend on a latent variable *z*, the MLE takes the following form:(1)$$\begin{array}{c} \max_{\theta }\log p_{\theta }\left(x\right)=\max_{\theta }\log\left(\int p_{\theta }\left(x|z\right)p\left(z\right)dz\right) \end{array}$$ For many latent models, the log evidence integral is unavailable in closed form or it is too complex to compute. A leading approach to handle such intractable cases is variational inference (VI). One of the most successful applications of VI in the deep neural network realm is the Variational Autoencoder (VAE).

### 2.1. Variational Autoencoder and the Kulback–Leibler Divergence

A Variational Autoencoder is a deep generative model that implements a probabilistic model and variational Bayesian inference. Introduced by Kingma and Welling [2], a VAE model is an autoencoder, designed to stochastically encode the input data into a constrained multivariate latent space (encoding), and then to reconstruct it as accurately as possible (decoding). To turn the intractable posterior inference into a solvable problem, we use a parametric inference model $q_{\phi }\left(z|x\right)$ which is also called an encoder. We optimize the variational parameters ϕ such that $q_{\phi }\left(z|x\right)∼p_{\theta }\left(z|x\right)$. The VAE loss function is composed of a “reconstruction term” (to ensure the decoded data are close to the original data) and a “regularisation term”. The goal of the regularisation term is to ensure that the distributions returned by the encoder are close to a standard normal distribution. That is expressed as the Kulback–Leibler divergence between the returned distribution and a standard Gaussian.

**Definition** **1.**
*Kulback–Leibler (KL) divergence [10,11]. For discrete probability distributions p and q, defined on the same probability space, the KL divergence from q to p is defined to be:*

(2)
$$D_{KL}\left(p||q\right)=\sum_{x}p\left(x\right)\log\left(\frac{p(x)}{q(x)}\right)$$



Since the true posterior $p_{\theta }\left(z|x\right)$ is intractable, we aim to approximate it with a Gaussian distribution $q_{\phi }\left(z|x\right)$, in the KL divergence sense. It follows that:(3)$$\begin{array}{cc} \log p_{\theta }\left(x\right) & =D_{KL}(q_{\phi }\left(z|x\right)||p_{\theta }\left(z|x\right))+\mathrm{ELBO} \end{array}$$

**Definition** **2.**
*Evidence Lower Bound (ELBO):*

(4)
$$\mathrm{ELBO}:=E_{z∼q_{\phi }\left(z|x\right)}\left[\log\left(p_{\theta }\left(z,x\right)\right)-\log\left(q_{\phi }\left(z|x\right)\right)\right]$$



We note that the KL divergence is non-negative, thus maximizing the ELBO results with the minimization of the KL divergence between $q_{\phi }\left(z|x\right)$ and the true posterior $p_{\theta }\left(z|x\right)$.

ELBO optimization is a well-known method that has been studied in depth, and is applicable in many models, especially in VAE [12]. Nevertheless, using the ELBO can give rise to some drawbacks. First, the ELBO is not always very tight, and maximizing the bound instead of the actual likelihood can lead to bias. Typically this leads to a simpler model $q_{\phi }$, which approximates the real posterior. Second, the $D_{KL}(q_{\phi }\left(z|x\right)||p_{\theta }\left(z|x\right))$ does not always lead to the best results—it tends to favor approximate distributions $q_{\phi }$ that underestimate the entropy of the true posterior (“zero-forcing”). Namely, $D_{KL}(q_{\phi }\left(z|x\right)||p_{\theta }\left(z|x\right))$ is infinite when $p_{\theta }\left(z|x\right)=0$ and $q_{\phi }\left(z|x\right)>0$. Therefore, the optimal variational distribution *q* will be 0 when $p_{\theta }\left(z|x\right)=0$. This “zero-forcing” behavior leads to degenerate solutions during optimization.

### 2.2. Rényi Divergence

One of the core parts of probabilistic models is the selection of the method for estimating the approximation of the distribution. In the previous section, we introduced Kulback–Leibler (KL) divergence. In this section, we will present the Rényi divergence (also known as α divergence), which measures the difference between two distributions *p* and *q*, and is defined by:(5)$$\begin{array}{cc} D_{\alpha }\left(p||q\right)= & \frac{1}{\alpha -1}\log\left(E_{p}\left[{\left(\frac{p(x)}{q(x)}\right)}^{\alpha -1}\right]\right)\\ = & \frac{1}{\alpha -1}\log\left(\sum_{x\in X}\frac{p{\left(x\right)}^{\alpha }}{q{\left(x\right)}^{\alpha -1}}\right) \end{array}$$ Rényi divergence was initially defined for α∈{α>0, α≠1}. The definition was extended to α=0,1,+∞ by continuity. There are certain α values for which Rényi divergence has a wider application than the others. Of particular interest are the values 0, $\frac{1}{2}$, 1, 2, and *∞*, presented in Table 1. We note that for α→1: $\lim_{\alpha \to 1}D_{\alpha }\left(p||q\right)=D_{KL}\left(p||q\right)$, the KL divergence is recovered.

#### 2.2.1. Selected Properties of Rényi Divergence

**Theorem** **1.**
*(Positivity): For any order α∈[0,∞]: $D_{\alpha }\left(p||q\right)\geq 0$, and*


$D_{\alpha }\left(p||q\right)=0\Leftrightarrow p=q$



**Theorem** **2.**
*(Convexity): For any order α∈[0,1] Rényi divergence is jointly convex in its arguments. That is, for any two pairs of probability distributions $(p_{0},q_{0}$) and $\left(p_{1},q_{1}\right)$, and any 0<λ<1:*

(6)
$$D_{\alpha }(\left(1-\lambda \right)p_{0}+\lambda p_{1}||\left(1-\lambda \right)q_{0}+\lambda q_{1})\leq \left(1-\lambda \right)D_{\alpha }(p_{0}\left|\right|q_{0})+\lambda D_{\alpha }(p_{1}\left|\right|q_{1})$$

*For any order α∈[0,∞] Rényi divergence is convex in its second argument. That is, for any probability distributions p, $q_{0}$ and $q_{1}$:*

(7)
$$D_{\alpha }(p||\left(1-\lambda \right)q_{0}+\lambda q_{1})\leq \left(1-\lambda \right)D_{\alpha }\left(p|\right|q_{0})+\lambda D_{\alpha }\left(p|\right|q_{1})$$



**Theorem** **3.**
*(Continuity in the Order): The Rényi divergence is continuous in α on $A=\{\alpha \in \left[0,\infty \right]|0\leq \alpha \leq 1\;\mathrm{or}\;D_{\alpha }\left(p||q)<\infty \right\}$.*


The definition of Rényi divergence was extended to α<0 as well. However, not all properties are preserved, and some are inverted. For example, Rényi divergence for negative orders is *non-positive* and *concave* in its first argument (cf. Figure 1). The extended definition of Rényi divergence to all α∈R has some interesting properties:

**Theorem** **4.**
*(Monotonicity) [3]: Rényi divergence, extended to negative α, is continuous and non-decreasing on $\alpha \in \{\alpha :-\infty \;<D_{\alpha }<+\infty \}$.*


**Lemma** **1.**
*The Skew Symmetry property:*


*For any α∈(−∞,∞),α∉{0,1}*

$$\begin{array}{cc} D_{\alpha }\left(p||q\right) & =\frac{\alpha }{1-\alpha }D_{1-\alpha }\left(q||p\right)\\ D_{-\infty }\left(p||q\right) & =-D_{\infty }\left(q||p\right) \end{array}$$


*For any α∈(−∞,∞),α∉{0,1}*

$$D_{\alpha }\left(p||q\right)\leq \frac{\alpha }{1-\alpha }D_{1-\alpha }\left(p||q\right)$$




**Definition** **3.**
*We will denote by $d_{\alpha }\left(p||q\right)$ the exponential:*

(8)
$$d_{\alpha }\left(p||q\right)=e^{D_{\alpha }\left(p||q\right)}={\left(\sum_{x\in X}\frac{p{\left(x\right)}^{\alpha }}{q{\left(x\right)}^{\alpha -1}}\right)}^{\frac{1}{\alpha -1}}$$



Figure 1 illustrates $d_{\alpha }$ and $D_{\alpha }$. One can see that $d_{\alpha }$ achieves high values very quickly. $D_{\alpha }\left(p||q\right)$ and $d_{\alpha }\left(p||q\right)$ are non-decreasing as functions of α, and:(9)$$d_{\alpha }\left(p||q\right)\leq d_{\infty }\left(p||q\right)=\underset{x\in X}{\sup}\left[\frac{p(x)}{q(x)}\right]$$

Many other properties described in [3,13].

#### 2.2.2. Rényi Divergence Variational Inference

To estimate the evidence $p_{\theta }\left(x\right)$, we employ a minimization approach using Rényi divergence between the variational distribution $q_{\phi }\left(z|x\right)$ and the true posterior distribution $p_{\theta }\left(z|x\right)$, where α is a selected positive value. Extending the posterior $p_{\theta }\left(z|x\right)$ and using Bayes’ theorem, we obtain:(10)$$\begin{array}{cc} D_{\alpha }(q_{\phi }\left(z|x\right)||p_{\theta }\left(z|x\right)) & =\frac{1}{\alpha -1}\log\left(E_{z∼q_{\phi }\left(z|x\right)}\left[{\left(\frac{q_{\phi }\left(z|x\right)}{p_{\theta }\left(z|x\right)}\right)}^{\alpha -1}\right]\right)\\ \; & =\frac{1}{\alpha -1}\log\left(E_{z∼q_{\phi }\left(z|x\right)}\left[{\left(\frac{p_{\theta }\left(z,x\right)}{q_{\phi }\left(z|x\right)\cdot p_{\theta }\left(x\right)}\right)}^{1-\alpha }\right]\right)\\ \; & =\log p_{\theta }\left(x\right)+\frac{1}{\alpha -1}\log\left(E_{z∼q_{\phi }\left(z|x\right)}\left[{\left(\frac{p_{\theta }\left(z,x\right)}{q_{\phi }\left(z|x\right)}\right)}^{1-\alpha }\right]\right) \end{array}$$ It follows that:(11)$$\begin{array}{c} \log p_{\theta }\left(x\right)=D_{\alpha }(q_{\phi }\left(z|x\right)||p_{\theta }\left(z|x\right))+{\mathrm{VR}}_{\alpha } \end{array}$$

**Definition** **4.**
*Variational Rényi (VR) bound [3]:*

(12)
$$\begin{array}{c} {\mathrm{VR}}_{\alpha }:=\frac{1}{1-\alpha }\log\left(E_{z∼q_{\phi }\left(z|x\right)}\left[{\left(\frac{p_{\theta }\left(z,x\right)}{q_{\phi }\left(z|x\right)}\right)}^{1-\alpha }\right]\right) \end{array}$$



The variational Rényi (VR) bound can be extended for α<0 as well. Since $D_{\alpha }\left(p||q\right)\geq 0$ for α≥0 and $D_{\alpha }\left(p||q\right)\leq 0$ for α≤0 (see Figure 1), then, for α≥0, ${\mathrm{VR}}_{\alpha }$ is a lower bound for $\log p_{\theta }\left(x\right)$, and for α≤0, ${\mathrm{VR}}_{\alpha }$ is an upper bound for $\log p_{\theta }\left(x\right)$.

### 2.3. χ Divergence

Similarly to the KL divergence and the Rényi divergence, one can use the $\chi^{2}$-divergence (or in general the $\chi^{n}$-divergence) and develop a bound of the log evidence [14].

**Definition** **5.**
*$\chi^{2}$-divergence:*

(13)
$$D_{\chi^{2}}\left(p||q\right)=E_{q}\left[{\left(\frac{p(x)}{q(x)}\right)}^{2}-1\right]$$



Now, our objective is to approximate the evidence $p_{\theta }\left(x\right)$ by using $\chi^{2}$-divergence between the true posterior $p_{\theta }\left(z|x\right)$ and $q_{\phi }\left(z|x\right)$.
(14)$$\begin{array}{cc} D_{\chi^{2}}(p_{\theta }\left(z|x\right)||q_{\phi }\left(z|x\right)) & =E_{z∼q_{\phi }\left(z|x\right)}\left[{\left(\frac{p_{\theta }\left(z|x\right)}{q_{\phi }\left(z|x\right)}\right)}^{2}-1\right]\\ \; & =E_{z∼q_{\phi }\left(z|x\right)}\left[{\left(\frac{p_{\theta }\left(z,x\right)}{p_{\theta }\left(x\right)q_{\phi }\left(z|x\right)}\right)}^{2}\right]-1\\ \; & =\frac{1}{p_{\theta }{\left(x\right)}^{2}}E_{z∼q_{\phi }\left(z|x\right)}\left[{\left(\frac{p_{\theta }\left(z,x\right)}{q_{\phi }\left(z|x\right)}\right)}^{2}\right]-1 \end{array}$$ After rearranging the equation we will obtain:(15)$$\begin{array}{c} E_{z∼q_{\phi }\left(z|x\right)}\left[{\left(\frac{p_{\theta }\left(z,x\right)}{q_{\phi }\left(z|x\right)}\right)}^{2}\right]=p_{\theta }{\left(x\right)}^{2}\left[1+D_{\chi^{2}}(p_{\theta }\left(z|x\right)||q_{\phi }\left(z|x\right))\right] \end{array}$$ Taking logarithms on both sides:(16)$$\begin{array}{c} \log\left(E_{z∼q_{\phi }\left(z|x\right)}\left[{\left(\frac{p_{\theta }\left(z,x\right)}{q_{\phi }\left(z|x\right)}\right)}^{2}\right]\right)=2\log p_{\theta }\left(x\right)+\log\left(\left[1+D_{\chi^{2}}(p_{\theta }\left(z|x\right)||q_{\phi }\left(z|x\right))\right]\right)\\ \log p_{\theta }\left(x\right)=\frac{1}{2}\log\left(E_{z∼q_{\phi }\left(z|x\right)}\left[{\left(\frac{p_{\theta }\left(z,x\right)}{q_{\phi }\left(z|x\right)}\right)}^{2}\right]\right)-\frac{1}{2}\log\left(\left[1+D_{\chi^{2}}(p_{\theta }\left(z|x\right)||q_{\phi }\left(z|x\right))\right]\right) \end{array}$$ By monotonicity of log and non-negativity of the $\chi^{2}$-divergence, this quantity is an upper bound of the log evidence:(17)$$\log p_{\theta }\left(x\right)\leq \frac{1}{2}\log\left(E_{z∼q_{\phi }\left(z|x\right)}\left[{\left(\frac{p_{\theta }\left(z,x\right)}{q_{\phi }\left(z|x\right)}\right)}^{2}\right]\right)$$

**Definition** **6.**
*χ upper bound (CUBO):*

(18)
$${\mathrm{CUBO}}_{2}:=\frac{1}{2}\log\left(E_{z∼q_{\phi }\left(z|x\right)}\left[{\left(\frac{p_{\theta }\left(z,x\right)}{q_{\phi }\left(z|x\right)}\right)}^{2}\right]\right)$$


(19)
$${\mathrm{CUBO}}_{n}:=\frac{1}{n}\log\left(E_{z∼q_{\phi }\left(z|x\right)}\left[{\left(\frac{p_{\theta }\left(z,x\right)}{q_{\phi }\left(z|x\right)}\right)}^{n}\right]\right)$$



Using $\chi^{n}$-divergence for general *n*, ${\mathrm{CUBO}}_{n}$ provides a family of bounds. We note the strong connection between the ${\mathrm{CUBO}}_{n}$ and the Rényi bound ${\mathrm{VR}}_{\alpha }$: when n=1−α, the VR bound is revealed.

**Theorem** **5.**
*(Sandwich Theorem [14]) For ${\mathrm{CUBO}}_{n}$ the following holds:*

*1*.
*∀n>1: $\mathrm{ELBO}\leq \log p_{\theta }\left(x\right)\leq {\mathrm{CUBO}}_{n}$.*
*2*.
*∀n>1: ${\mathrm{CUBO}}_{n}$ is a non-decreasing function of the n order χ-divergence.*
*3*.
*$\lim_{n\to 0}{\mathrm{CUBO}}_{n}=\mathrm{ELBO}$.*



Using Theorem 5, one can estimate $\log p_{\theta }\left(x\right)$ with both upper and lower bounds, which may provide a better approximation for the log evidence.

The χ upper bound has many advantages: It is a black-box inference algorithm in that it does not need model-specific derivations and it is easy to apply to a wide class of models. In addition, it is useful when the KL divergence is not a good objective, and it is guaranteed to converge [14].

### 2.4. Monte Carlo Approximation

So far, we have discussed KL divergence, Rényi divergence, and χ divergence, and have demonstrated how each of these measurements can be used to construct a bound for the log evidence. However, calculating these bounds is computationally intractable, due to the stochastic nature of the latent space and the exponential number of random variables. In real-world situations, where datasets are typically limited and contain a finite number of data points, empirical estimations become necessary. A popular method for estimating these bounds is the Monte Carlo (MC) approximation [15,16]. Typically, the MC method involves random sampling from certain probability distributions.

The Monte Carlo (MC) approximation of the Kullback–Leibler (KL) divergence is unbiased, guaranteeing the convergence of the optimization process for the Evidence Lower Bound (ELBO). However, the MC approximation for the Rényi bound introduces bias, leading to an underestimation of the true expectation. In the case of positive values of α, this implies a relatively looser bound, but it should still be effective. On the other hand, for negative values of α, this becomes a significant issue as it underestimates an upper bound. More precisely, the MC approximation for the Rényi bound is:(20)$${\hat{\mathrm{VR}}}_{\alpha }=\frac{1}{1-\alpha }\log\left(\frac{1}{K}\sum_{i=1}^{K}{\left(\frac{p_{\theta }\left(z_{i},x\right)}{q_{\phi }\left(z_{i}|x\right)}\right)}^{1-\alpha }\right)$$ For this to be unbiased, the expectation should be equal to the true value,
(21)$$\begin{array}{c} E_{q_{\phi }}\left[{\hat{\mathrm{VR}}}_{\alpha }\right]=\frac{1}{1-\alpha }E_{q_{\phi }}\left[\log\left(\frac{1}{K}\sum_{i=1}^{K}{\left(\frac{p_{\theta }\left(z_{i},x\right)}{q_{\phi }\left(z_{i}|x\right)}\right)}^{1-\alpha }\right)\right] \end{array}$$ By Jensen’s inequality:(22)$$\begin{array}{cc} \;\;\;\;\;\;\;\;\;\;\;\;\;\;\; & \leq \frac{1}{1-\alpha }\log\left(E_{q_{\phi }}\left[\frac{1}{K}\sum_{i=1}^{K}{\left(\frac{p_{\theta }\left(z_{i},x\right)}{q_{\phi }\left(z_{i}|x\right)}\right)}^{1-\alpha }\right]\right)\\ \; & ={\mathrm{VR}}_{\alpha } \end{array}$$ Thus, the approximation is actually an underestimate of the true bound. This characteristic was also discussed in [3], where the authors suggested improving the approximation quality by using more samples and using negative α values to improve the accuracy, at the cost of losing the upper-bound guarantee.

Other papers have suggested different approaches to keep the upper bounding property intact [8,14,17]. Of particular interest is the generic χ upper bound, ${\mathrm{CUBO}}_{n}$, which also suffers from the same problem of biased estimation using MC approximation. In [14], the authors suggested an approach to avoid the biased approximation, by exponentiation:(23)$$L=e^{n\cdot {\mathrm{CUBO}}_{n}}$$ Applying MC approximation to L provides an unbiased upper bound. However, this change affects the variance of the gradients, which may damage the quality of the approximation result. It may result in high variance estimates and requires a large number of samples in order to serve as a reliable upper bound [18].

## 3. Improved VR Bound and Upper–Lower Bound Optimization

### 3.1. Variational Rényi Log Upper Bound (VRLU)

We suggest a different approach for estimating the upper bound while preserving the upper bound property. Consider the following inequalities:(24)$$1-\frac{1}{x}\leq \log x\leq x-1$$ where equality holds on both sides if and only if x=1.

**Definition** **7.**
*Variational Rényi Log Upper bound (VRLU):*

(25)
$${\mathrm{VRLU}}_{\alpha }:=\frac{1}{1-\alpha }\left(E_{z∼q_{\phi }\left(z|x\right)}\left[{\left(\frac{p_{\theta }\left(z,x\right)}{q_{\phi }\left(z|x\right)}\right)}^{1-\alpha }\right]-1\right)$$


(26)
$${\hat{\mathrm{VRLU}}}_{\alpha }:=\frac{1}{1-\alpha }\left(\left(\frac{1}{K}\sum_{i=1}^{K}{\left(\frac{p_{\theta }\left(z_{i},x\right)}{q_{\phi }\left(z_{i}|x\right)}\right)}^{1-\alpha }\right)-1\right)$$



For negative α, ${\hat{\mathrm{VRLU}}}_{\alpha }$ is an estimation of the Rényi upper bound, and an upper bound of the log evidence: (27)$$\begin{array}{cc} E_{q_{\phi }}\left[{\hat{\mathrm{VRLU}}}_{\alpha }\right]\; & =E_{q_{\phi }}\left[\frac{1}{1-\alpha }\left(\left(\frac{1}{K}\sum_{i=1}^{K}{\left(\frac{p_{\theta }\left(z_{i},x\right)}{q_{\phi }\left(z_{i}|x\right)}\right)}^{1-\alpha }\right)-1\right)\right]\\ \; & \geq \frac{1}{1-\alpha }\log\left(E_{q_{\phi }}\left[\frac{1}{K}\sum_{i=1}^{K}{\left(\frac{p_{\theta }\left(z_{i},x\right)}{q_{\phi }\left(z_{i}|x\right)}\right)}^{1-\alpha }\right]\right)\\ \; & =\frac{1}{1-\alpha }\log\left(E_{q_{\phi }}\left[{\left(\frac{p_{\theta }\left(z_{i},x\right)}{q_{\phi }\left(z_{i}|x\right)}\right)}^{1-\alpha }\right]\right) \end{array}$$ Note that the inequalities in (Equation 24) become tighter as the argument of the log is closer to 1. In the Rényi bound approximation (Equation 20), this argument is $1/k\sum {\left(p_{\theta }\left(z,x\right)/q_{\phi }\left(z|x\right)\right)}^{1-\alpha }$. Thus, the approximation becomes tighter as the variational distribution, $q_{\phi }$, is getting closer to the true distribution $p_{\theta }$ (the lower the divergence, the tighter the approximation), which is exactly the goal of the optimization.

We evaluated the bias of MC approximations for both bounds, ${\mathrm{VR}}_{\alpha }$ and ${\mathrm{VRLU}}_{\alpha }$, over a range of negative α values. To this end, we fixed the distributions *p* and *q* to both be Gaussian: p∼N(0,1), q∼N(1.5,1). The bounds ${\mathrm{VR}}_{\alpha }$ and ${\mathrm{VRLU}}_{\alpha }$ were estimated using the MC approximation (cf. (Equation 26) and (Equation 20)) and we evaluated the quality of the approximation for different values of MC samples, denoted by *K*.

Figure 2 shows the empirical results. We can see that the MC approximations for ${\mathrm{VR}}_{\alpha }$ are biased and get better as the sample size *K* increases. Furthermore, the bias results in an underestimation of ${\mathrm{VR}}_{\alpha }$ for α≤0, which makes it unattractive to be used as an upper bound at the negative α range. On the other hand, the MC approximation for ${\mathrm{VRLU}}_{\alpha }$ preserves the upper bound property and has a relatively low variance. As a result, ${\mathrm{VRLU}}_{\alpha }$ is a more suitable choice as an upper bound for negative α and may be used as an objective for risk minimization.

Figure 3 presents the comparison between ${\mathrm{VR}}_{\alpha }\left(p||q\right)$ and ${\mathrm{VRLU}}_{\alpha }\left(p||q\right)$ over different values of *q*. To this end, we fixed p∼N(1,2) and set q∼N(μ,2) while varying μ in the range [−5,10]. We can see that as closer *q* is to *p*, both ${\mathrm{VR}}_{\alpha }\left(p||q\right)$ and ${\mathrm{VRLU}}_{\alpha }\left(p||q\right)$ values are decreasing, and for p=q, ${\mathrm{VR}}_{\alpha }\left(p||q\right)={\mathrm{VRLU}}_{\alpha }\left(p||q\right)=0$ for all α values. Furthermore, as α is farther away from 0, the steeper the graph becomes.

In conclusion, we empirically evaluated the ${\mathrm{VRLU}}_{\alpha }$ upper bound and matched it against the ${\mathrm{VR}}_{\alpha }$ upper bound, for varying values of negative α. The divergence curve of the ${\mathrm{VRLU}}_{\alpha }$ upper bound is steeper than the ${\mathrm{VR}}_{\alpha }$ upper bound, and the variance is much lower, suggesting a higher convergence as the variational distribution is getting closer to the true posterior.

### 3.2. Upper–Lower Bound Optimization

Using the new upper bound, ${\mathrm{VRLU}}_{\alpha }$, we devised ${\mathrm{VRS}}_{\alpha_{+},\alpha_{-}}$; a (sandwiched) upper–lower bound variational inference algorithm for jointly minimizing the Rényi upper and lower bounds. ${\mathrm{VRS}}_{\alpha_{+},\alpha_{-}}$ combined both the upper and lower Rényi bounds, where the lower bound ${\mathrm{VR}}_{\alpha }$ is computed as in Equation (Equation 20) for a constant positive α, and the upper bound ${\mathrm{VRLU}}_{\alpha }$ is computed as in Equation (Equation 26) for a constant negative α. The overall ${\mathrm{VRS}}_{\alpha_{+},\alpha_{-}}$ loss is the average of both terms, i.e.,
(28)$${\mathrm{VRS}}_{\alpha_{+},\alpha_{-}}:=\frac{1}{2}\cdot \left({\mathrm{VRLU}}_{\alpha_{-}}+{\mathrm{VR}}_{\alpha_{+}}\right)$$
(29)$${\hat{\mathrm{VRS}}}_{\alpha_{+},\alpha_{-}}=\frac{1}{2}\cdot \left({\hat{\mathrm{VRLU}}}_{\alpha_{-}}+{\hat{\mathrm{VR}}}_{\alpha_{+}}\right)$$ Since ${\mathrm{VR}}_{\alpha_{+}}\leq \log p_{\theta }\left(x\right)\leq {\mathrm{VR}}_{\alpha_{-}}\leq {\mathrm{VRLU}}_{\alpha_{-}}$, the ${\mathrm{VRS}}_{\alpha_{+},\alpha_{-}}$ loss provides a useful estimate for the log-likelihood of the evidence.

### 3.3. Probability Approximation

Recall that our objective is to develop a probabilistic model, denoted as $p_{\theta }$, that effectively captures and explains the underlying data. In variational inference (VI), we tackle an optimization problem that seeks to find a simpler distribution that closely approximates the original data distribution, also known as the evidence. In this section, we will inspect the approximate distribution, denoted as $\hat{p_{\theta }}$, that minimizes the divergence $d_{\alpha }(\hat{p_{\theta }}\left|\right|p_{\theta })$. Our aim is to find an approximation that accurately represents the true data distribution.

We will evaluate two methods of approximating $p_{\theta }$. One using VR bound:(30)$$\hat{p_{\theta }}\left(x\right)=\frac{e^{{\mathrm{VR}}_{\alpha }\left(x\right)}}{\sum_{x\in X}e^{{\mathrm{VR}}_{\alpha }\left(x\right)}}$$
and one using our VRS method:(31)$$\hat{p_{\theta }}\left(x\right)=\frac{e^{{\mathrm{VRS}}_{\alpha_{+},\alpha_{-}}\left(x\right)}}{\sum_{x\in X}e^{{\mathrm{VRS}}_{\alpha_{+},\alpha_{-}}\left(x\right)}}$$ We notice that for both estimators, $\hat{p_{\theta }}$ is indeed a probability. Given that both ${\mathrm{VR}}_{\alpha }$ and ${\mathrm{VRS}}_{\alpha_{+},\alpha_{-}}$ estimate the log evidence, we will use the exponent of these estimates to approximate $p_{\theta }$.

Let us denote $\alpha_{+}>0$. Using Equation (Equation 11),
(32)$$\begin{array}{cc} e^{{\mathrm{VR}}_{\alpha_{+}}\left(x\right)}\; & =e^{\log p_{\theta }\left(x\right)-D_{\alpha_{+}}(q_{\phi }\left(z|x\right)||p_{\theta }\left(z|x\right))}\\ \; & =\frac{e^{\log p_{\theta }\left(x\right)}}{e^{D_{\alpha_{+}}(q_{\phi }\left(z|x\right)||p_{\theta }\left(z|x\right))}}\\ \; & =\frac{p_{\theta }\left(x\right)}{d_{\alpha_{+}}(q_{\phi }\left(z|x\right)||p_{\theta }\left(z|x\right))} \end{array}$$ Let us denote $\alpha_{-}<0$.
(33)$$\begin{array}{cc} e^{{\mathrm{VR}}_{\alpha_{-}}\left(x\right)}\; & =e^{\log p_{\theta }\left(x\right)-D_{\alpha_{-}}(q_{\phi }\left(z|x\right)||p_{\theta }\left(z|x\right))}\\ \; & =e^{\log p_{\theta }\left(x\right)}e^{-D_{\alpha_{-}}(q_{\phi }\left(z|x\right)||p_{\theta }\left(z|x\right))}\\ \; & =\frac{p_{\theta }\left(x\right)}{d_{\alpha_{-}}(q_{\phi }\left(z|x\right)||p_{\theta }\left(z|x\right))} \end{array}$$ Using both upper and lower bounds we will find that:(34)$$\begin{array}{cc} e^{{\mathrm{VRS}}_{\alpha_{+},\alpha_{-}}\left(x\right)} & =e^{\frac{1}{2}\left({\mathrm{VR}}_{\alpha_{+}}\left(x\right)+{\mathrm{VR}}_{\alpha_{-}}\left(x\right)\right)}\\ \; & ={\left(e^{{\mathrm{VR}}_{\alpha_{+}}\left(x\right)}e^{{\mathrm{VR}}_{\alpha_{-}}\left(x\right)}\right)}^{\frac{1}{2}}\\ \; & =\sqrt{\frac{{p_{\theta }\left(x\right)}^{2}}{d_{\alpha_{+}}(q_{\phi }\left(z|x\right)||p_{\theta }\left(z|x\right))d_{\alpha_{-}}(q_{\phi }\left(z|x\right)||p_{\theta }\left(z|x\right))}}\\ \; & =p_{\theta }\left(x\right)\sqrt{\frac{1}{d_{\alpha_{+}}(q_{\phi }\left(z|x\right)||p_{\theta }\left(z|x\right))d_{\alpha_{-}}(q_{\phi }\left(z|x\right)||p_{\theta }\left(z|x\right))}} \end{array}$$ We will define multiplication factors for both our approximations as follows:(35)$${\mathrm{VRS}}_{MF}:=\frac{1}{\sqrt{d_{\alpha_{+}}(q_{\phi }\left(z|x\right)||p_{\theta }\left(z|x\right))d_{\alpha_{-}}(q_{\phi }\left(z|x\right)||p_{\theta }\left(z|x\right))}}$$
(36)$${\mathrm{VR}}_{MF}:=\frac{1}{d_{\alpha_{+}}(q_{\phi }\left(z|x\right)||p_{\theta }\left(z|x\right))}$$ Note that $e^{{\mathrm{VRS}}_{\alpha_{+},\alpha_{-}}\left(x\right)}=p_{\theta }\left(x\right)\cdot {\mathrm{VRS}}_{MF}$ and $e^{{\mathrm{VR}}_{\alpha }\left(x\right)}=p_{\theta }\left(x\right)\cdot {\mathrm{VR}}_{MF}$. Thus, our goal is to achieve a multiplication factor as close to one as possible. We examine these values using the fixed distribution p∼N(0,2), and distribution q∼N(μ,2), where −3<μ<3. When μ=0, p=q. We used different $\alpha_{+}$ and $\alpha_{-}$ values. The results are presented in Figure 4.

We can see that for every $\alpha_{+}$, ${\mathrm{VRS}}_{MF}$ is closer to one for all different $\alpha_{-}$ values compared to ${\mathrm{VR}}_{MF}$ with the same $\alpha_{+}$ value. In addition, when $\alpha_{-}$ and $\alpha_{+}$ are symmetric around zero, the multiplication factor of ${\mathrm{VRS}}_{\alpha_{+},\alpha_{-}}$ is closest to one. This indicates that the $\hat{p_{\theta }}\left(x\right)$ approximation calculated using ${\mathrm{VRS}}_{\alpha_{+},\alpha_{-}}$ is more accurate among the two methods.

## 4. Multiple-Source Adaptation (MSA)

In statistical learning, there are numerous settings that require an accurate estimation of the data distribution to find effective solutions. One such task is known as domain adaptation. In the preceding section, we introduced VRS as an enhanced method to obtain accurate approximations of the data distribution. In this section, we will apply these estimated distributions to the domain adaptation objective, thus demonstrating the effectiveness and practicality of the VRS method to yield accurate solutions.

Domain adaptation is a scenario where we aim to train a classifier on one dataset (referred to as the source domain) for which labels or annotations are available and achieve good performance on another dataset (referred to as the target domain) for which labels or annotations are not available. A common example of a domain adaptation application is spam filtering, where a model trained on one user’s emails (the source domain) is adapted and used to filter spam for a different user who receives distinct emails (the target domain).

In this work, our focus is on the Multi-Source Domain Adaptation (MSA) problem, where there are multiple source domains available in addition to only one target domain. The target domain can be considered as either an exact mixture of the source domains, or it might be well approximated by such a mixture. The goal is to leverage the information provided by the source domains to improve the performance on the target domain, where annotations or labels are not available.

In many real-world scenarios, the learner may not have access to all of the source data at once, due to privacy or storage constraints. Therefore, the learner cannot simply combine all of the source data together to train a predictor. A possible solution to this problem is the Mixture of Experts (MOE) approach. MOE is an ensemble learning technique that involves training multiple experts on different sub-tasks of a predictive modeling problem. Each expert concentrates on a specific part of the modeling problem space. A gating network then combines the outputs of the various experts. In the domain adaptation problem, this concept can be applied by modeling the domain relationship with an MOE approach.

The MSA problem was theoretically analyzed by Mansour, Mohri, and Rostamizadeh in [19]. In their paper, the authors presented the domain adaptation problem setup and proved that for any target domain, there exists a hypothesis, referred to as the distribution weighted combining rule, which is capable of achieving a low error rate with respect to the target domain. However, it should be noted that the authors did not provide a method for determining or learning the aforementioned hypothesis.

In the paper by Hoffman, Mohri and Zhang [9], the authors extended the definition of the weighted combination rule to solve probabilistic models as well, using cross-entropy loss. Additionally, the authors introduced an iterative algorithm based on Difference of Convex (DC) programming, that constructs the weighted combination rule. Nonetheless, the algorithm proposed in the paper assumes either prior knowledge of the probabilities associated with the data samples or relies on accurate estimates of these probabilities. The authors evaluated the performance of their model by employing the Rényi divergence, which quantifies the discrepancy between the true distribution and the approximated distribution. As a result, the effectiveness of their model is contingent upon the accuracy of the probability approximations as well.

In order to circumvent the need for good estimates of the data distribution, Cortes et al. [20] proposed a discriminative technique using an estimate of the conditional probabilities p(i|x) for each source domain i∈{1,...,k} (that is, the probability that an instance *x* belongs to source *i*). To this end, they had to modify the DC algorithm proposed in [9], in order to adapt to their new distribution calculation.

In this study, we will build upon the algorithm introduced by Hoffman, Mohri, and Zhang [9], and enhance it with a refined approximation of the source distribution via variational inference.

### 4.1. MSA Problem Setup

We refer to a probability model where there is a distribution over the input space *X*. Each data point x∈X has a corresponding label y∈Y, where *Y* denotes the space of labels. Our objective function describes the correspondence between the data point and its label f:X→Y. We will focus on the adaptation problem with *k* source domains and a single target domain. For each domain i∈{1,...,k}, we have a source distribution $p_{i}$ and corresponding hypotheses $h_{i}\left(x,y\right)\to \left[0,1\right]$. More precisely, $h_{i}$ returns the probability that f(x)=y.

**Definition** **8.**
*Let L:R→R be a loss function penalizing errors with respect to f. The loss of hypothesis h with respect to the objective function f and a distribution p is denoted by L(h,p,f) and defined as:*

(37)
$$L\left(h,p,f\right):=E_{p}\left[L\left(h,f\right)\right]=\sum_{x\in X}p\left(x\right)L\left(h(x,f(x))\right)$$



For simplicity, we will denote $L\left(h(x,f(x))\right)$ as L(h,f) throughout this paper. We will assume that the following properties hold for the loss function *L*:*L* is non-negative: L(x)≥0∀x∈R*L* is convex.*L* is bounded: ∃M≥0 s.t.∀x∈R:L(x)≤M.*L* is continuous in both arguments.*L* is symmetric.

**Proposition** **1.**
*For each domain i, the hypothesis $h_{i}$ is a relatively accurate predictor for domain i with the distribution $p_{i}$; i.e., there exists ϵ>0 such that:*

(38)
$$\forall i\in \left\{1,...,k\right\},\;\;L\left(h_{i},p_{i},f\right)\leq \epsilon$$



**Proposition** **2.**
*We will denote the simplex: $\Delta =\{\lambda :\lambda_{i}\geq 0\wedge \sum_{i=1}^{k}\lambda_{i}=1\}$. The distribution of the target domain $p_{T}$ is assumed to be a mixture of the k source distributions $p_{1},...,p_{k}$, that is:*

(39)
$$p_{T}\left(x\right)=\sum_{i=1}^{k}\lambda_{i}p_{i}\left(x\right)\;\;(\mathrm{for}\;\lambda \in \Delta )$$



### 4.2. Existence of a Good Hypothesis

The goal of solving the MSA problem is to establish a good predictor (a good predictor: a predictor that provides a small error with respect to the target domain) for the target domain, given the source domain’s predictors. A common assumption is that there exists some relationship between the target domain and the distributions of the source domains (See Proposition 2). It can be demonstrated that conventional convex combinations of source predictors may yield suboptimal results in certain scenarios. In particular, studies have indicated that even if the source predictors possess zero loss, no convex combination can attain a loss lower than a specific constant for a uniform mixture of the source distributions.

Alternately, Mansour, Mohri, and Rostamizadeh [19] proposed a distribution-weighted solution and defined the distribution-weighted combination hypothesis for a regression model. Hoffman and Mohri [9] extended the distribution-weighted combination hypothesis to a probabilistic model, as follows:

**Definition** **9.**
*Distribution-weighted combination hypothesis.*

*For any λ∈Δ,η>0 and (x,y)∈X×Y:*

(40)
$$h_{w}^{\eta }\left(x,y\right)=\sum_{i=1}^{k}\frac{w_{i}p_{i}\left(x\right)+\eta \frac{U(x)}{k}}{\sum_{j=1}^{k}\left(w_{j}p_{j}\left(x\right)\right)+\eta U\left(x\right)}h_{i}\left(x,y\right)$$

*where U(x) is the uniform distribution over X.*


In the probabilistic model case, we will use *L* as the binary cross entropy loss:(41)L(h,f)=−logh(x,f(x))
which maintain all of the required properties stated in Section 4.1.

**Theorem** **6.**
*For any target function $f\in \{f:\forall i\in \left\{1,...,k\right\},L\left(h_{i},p_{i},f\right)\leq \epsilon \}$ and for any δ>0, there exist η>0 and w∈Δ such that $L(h_{w}^{\eta },p_{\lambda },f)\leq \epsilon +\delta$ for any mixture parameter λ.*


The proof of Theorem 6 is detailed in [19]. From this Theorem, it can be inferred that for any fixed target function *f*, the distribution-weighted combination hypothesis is a good hypothesis for the target domain.

### 4.3. A Good Hypothesis with Estimated Probabilities

On closer inspection of Definition 9, it is evident that constructing $h_{w}^{\eta }$ requires access to the distributions of all domains, represented by $p_{i}\left(x\right)\;\;\forall i\in 1,...,k$. Yet, in practical settings, the true distributions $p_{i}$ may not be directly available to the learner. Instead, the learner relies on estimates ${\hat{p}}_{i}$ derived from the available data. Thus, addressing the application of domain adaptation becomes essential for real-world scenarios where the true distributions remain unknown.

Our objective is to minimize the value of $L(h_{i},\hat{p_{i}},f)$. To accomplish this, we will develop an upper bound for this loss function (similar to previous research [9,21]). By doing so, we can examine the impact of utilizing estimated distributions $\hat{p_{i}}$ on the efficacy of our model and gain insights into the application of domain adaptation in real-world scenarios. First, let us recall Holder’s inequality:

**Theorem** **7.**
*Holder’s inequality: For any s and t in the open interval (1,∞) with $\frac{1}{s}+\frac{1}{t}=1$, and for $\left\{x_{j}\right\}$ and $\left\{y_{j}\right\}$
j∈{1,...,k} be certain sets of real numbers, we have:*

(42)
$$\sum_{j=1}^{n}\left|x_{j}y_{j}\right|\leq {\left(\sum_{j=1}^{n}{\left|x_{j}\right|}^{s}\right)}^{\frac{1}{s}}{\left(\sum_{j=1}^{n}{\left|y_{j}\right|}^{t}\right)}^{\frac{1}{t}}$$



**Corollary** **1.**
*Let $\hat{p_{i}}$ be an estimation of the original domain distribution $p_{i}$. The following inequality holds for any α>1:*

(43)
$$L\left(h_{i},\hat{p_{i}},f\right)\leq {\left(d_{\alpha }(\hat{p_{i}}\left|\right|p_{i})\epsilon \right)}^{\frac{\alpha -1}{\alpha }}M^{\frac{1}{\alpha }}$$



**Proof of Corollary** **1.**For any hypothesis *h* and any distributions *p*, *q*, and for any α>1, the following holds (the proof is based on a similar corollary proven in [9]):
$$\begin{array}{ccc} L(h,q,f)\; & =\sum_{x\in X}q\left(x\right)L\left(h,f\right)\\ \; & =\sum_{x\in X}\left(\frac{q(x)}{p{\left(x\right)}^{\frac{\alpha -1}{\alpha }}}\right)p{\left(x\right)}^{\frac{\alpha -1}{\alpha }}L\left(h,f\right)\\ \; & \leq {\left(\sum_{x\in X}\frac{q{\left(x\right)}^{\alpha }}{p{\left(x\right)}^{\alpha -1}}\right)}^{\frac{1}{\alpha }}{\left(\sum_{x\in X}p\left(x\right)L{\left(h,f\right)}^{\frac{\alpha }{\alpha -1}}\right)}^{\frac{\alpha -1}{\alpha }}\; & \mathrm{By}\mathrm{Holder}’s\\ \; & \; & \mathrm{inequality}\mathrm{for}\\ \; & \; & s=\alpha ,\mathrm{and}\\ \; & \; & t=\frac{\alpha }{\alpha -1} \end{array}$$
$$\begin{array}{ccc} \;\;\;\;\;\;\;\;\;\;\;\;\;\;\;\;\;\;\; & ={\left({\left(\sum_{x\in X}\frac{q{\left(x\right)}^{\alpha }}{p{\left(x\right)}^{\alpha -1}}\right)}^{\frac{1}{\alpha -1}}\right)}^{\frac{\alpha -1}{\alpha }}{\left(\sum_{x\in X}p\left(x\right)L\left(h,f\right)L{\left(h,f\right)}^{\frac{1}{\alpha -1}}\right)}^{\frac{\alpha -1}{\alpha }}\\ \; & ={\left(d_{\alpha }\left(q||p\right)\right)}^{\frac{\alpha -1}{\alpha }}{\left(\sum_{x\in X}p\left(x\right)L\left(h,f\right)L{\left(h,f\right)}^{\frac{1}{\alpha -1}}\right)}^{\frac{\alpha -1}{\alpha }}\; & \mathrm{By}\mathrm{Definition}\;3\\ \; & \leq {\left(d_{\alpha }\left(q||p\right)\right)}^{\frac{\alpha -1}{\alpha }}{\left(\sum_{x\in X}p\left(x\right)L\left(h,f\right)M^{\frac{1}{\alpha -1}}\right)}^{\frac{\alpha -1}{\alpha }}\; & \mathrm{Since}M\geq |L(h,f)|\\ \; & \; & \mathrm{and}\frac{1}{\alpha -1}>0\\ \; & ={\left(d_{\alpha }\left(q||p\right)L\left(h,p,f\right)\right)}^{\frac{\alpha -1}{\alpha }}M^{\frac{1}{\alpha }}\; & \end{array}$$For each i∈{1,...,k}, by setting $p:=p_{i},q:=\hat{p_{i}}$ and $h:=h_{i}$, we will find that:
$$\begin{array}{ccc} L(h_{i},\hat{p_{i}},f)\; & \leq {\left(d_{\alpha }(\hat{p_{i}}\left|\right|p_{i})L\left(h_{i},p_{i},f\right)\right)}^{\frac{\alpha -1}{\alpha }}M^{\frac{1}{\alpha }}\; & \\ \; & \leq {\left(d_{\alpha }(\hat{p_{i}}\left|\right|p_{i})\epsilon \right)}^{\frac{\alpha -1}{\alpha }}M^{\frac{1}{\alpha }}\; & \mathrm{By}\mathrm{Proposition}1\; \end{array}$$□

Corollary 1 provides us an upper bound of the loss using the estimated distributions $\hat{p_{i}}$. When $\hat{p_{i}}\to p_{i}$, $d_{\alpha }(\hat{p_{i}}\left|\right|p_{i})\to 1$ and we will remain with $\epsilon^{\frac{\alpha -1}{\alpha }}M^{\frac{1}{\alpha }}$. We will set M=1, since we use the loss function $L\left(h,f\right)=-\log\left(h(x,f(x))\right)$ as the cross-entropy loss (log-loss). Thus, when $\hat{p_{i}}\to p_{i}$, ${\left(d_{\alpha }(\hat{p_{i}}\left|\right|p_{i})\epsilon \right)}^{\frac{\alpha -1}{\alpha }}M^{\frac{1}{\alpha }}\to \epsilon^{\frac{\alpha -1}{\alpha }}$.

By performing the aforementioned calculation with α<1, it is possible to derive a lower bound for $L(h_{i},{\hat{p}}_{i},f)$. This lower bound serves as a confirmation that the utilization of approximated probabilities does not lead to significant errors. For instance, if the lower bound exhibits a considerably large value, it indicates that our approximation is inadequate. Conversely, if the lower bound demonstrates a small value, it signifies the effectiveness of our approximation. Moreover, by employing both upper and lower bounds, we can obtain a more precise estimation of the loss.

**Theorem** **8.**
*Generalization of Holder’s inequality [22]: Let 0<s<1 and t∈R with $\frac{1}{s}+\frac{1}{t}=1$, and for $\left\{x_{j}\right\}$ and $\left\{y_{j}\right\}$
j∈{1,...,n} be certain sets of real numbers, we have:*

(44)
$$\sum_{j=1}^{n}\left|x_{j}y_{j}\right|\geq {\left(\sum_{j=1}^{n}{\left|x_{j}\right|}^{s}\right)}^{\frac{1}{s}}{\left(\sum_{j=1}^{n}{\left|y_{j}\right|}^{t}\right)}^{\frac{1}{t}}$$



**Corollary** **2.**
*Let $\hat{p_{i}}$ be an estimation of the original domain distribution $p_{i}$. The following inequality holds for any α<1:*

(45)
$$L\left(h_{i},\hat{p_{i}},f\right)\geq {\left(d_{\alpha }(\hat{p_{i}}\left|\right|p_{i})\right)}^{\frac{\alpha -1}{\alpha }}\psi$$

*where $\psi ={\left(\sum_{x\in X}p_{i}\left(x\right)L{\left(h_{i},f\right)}^{\frac{\alpha }{\alpha -1}}\right)}^{\frac{\alpha -1}{\alpha }}$*


**Proof of Corollary** **2.**First, we will prove for 0<α<1, and then for α<0. Let us set 0<α<1, s=α and $t=\frac{\alpha }{\alpha -1}$. For any hypothesis *h* and any distributions *p*, *q*, the following holds:
$$\begin{array}{ccc} L(h,q,f)\; & =\sum_{x\in X}q\left(x\right)L\left(h,f\right)\; & \\ \; & =\sum_{x\in X}\left(\frac{q(x)}{p{\left(x\right)}^{\frac{\alpha -1}{\alpha }}}\right)p{\left(x\right)}^{\frac{\alpha -1}{\alpha }}L\left(h,f\right) & \\ \; & \geq {\left(\sum_{x\in X}\frac{q{\left(x\right)}^{\alpha }}{p{\left(x\right)}^{\alpha -1}}\right)}^{\frac{1}{\alpha }}{\left(\sum_{x\in X}p\left(x\right)L{\left(h,f\right)}^{\frac{\alpha }{\alpha -1}}\right)}^{\frac{\alpha -1}{\alpha }}\; & \mathrm{By}\mathrm{the}\mathrm{generalization}\mathrm{of}\\ \; & \; & \mathrm{Holder}’s\mathrm{inequality}\mathrm{for}\\ \; & \; & s=\alpha ,t=\frac{\alpha }{\alpha -1}\\ \; & ={\left({\left(\sum_{x\in X}\frac{q{\left(x\right)}^{\alpha }}{p{\left(x\right)}^{\alpha -1}}\right)}^{\frac{1}{\alpha -1}}\right)}^{\frac{\alpha -1}{\alpha }}{\left(\sum_{x\in X}p\left(x\right)L{\left(h,f\right)}^{\frac{\alpha }{\alpha -1}}\right)}^{\frac{\alpha -1}{\alpha }}\; & \\ \; & ={\left(d_{\alpha }\left(q||p\right)\right)}^{\frac{\alpha -1}{\alpha }}{\left(\sum_{x\in X}p\left(x\right)L{\left(h,f\right)}^{\frac{\alpha }{\alpha -1}}\right)}^{\frac{\alpha -1}{\alpha }}\; & \mathrm{By}\mathrm{Definition}\;3 \end{array}$$Next, let us set α<0, t=α and $s=\frac{\alpha }{\alpha -1}$ (notice that α<0→0<s<1).For any hypothesis *h* and any distributions *p*, *q*, the following holds:
$$\begin{array}{ccc} L(h,q,f)\; & =\sum_{x\in X}q\left(x\right)L\left(h,f\right)\; & \\ \; & =\sum_{x\in X}\left(\frac{q(x)}{p{\left(x\right)}^{\frac{\alpha -1}{\alpha }}}\right)p{\left(x\right)}^{\frac{\alpha -1}{\alpha }}L\left(h,f\right) & \\ \; & =\sum_{x\in X}p{\left(x\right)}^{\frac{\alpha -1}{\alpha }}L\left(h,f\right)\left(\frac{q(x)}{p{\left(x\right)}^{\frac{\alpha -1}{\alpha }}}\right) & \\ \; & \geq {\left(\sum_{x\in X}p\left(x\right)L{\left(h,f\right)}^{\frac{\alpha }{\alpha -1}}\right)}^{\frac{\alpha -1}{\alpha }}{\left(\sum_{x\in X}\frac{q{\left(x\right)}^{\alpha }}{p{\left(x\right)}^{\alpha -1}}\right)}^{\frac{1}{\alpha }}\; & \mathrm{By}\mathrm{the}\mathrm{generalization}\mathrm{of}\\ \; & \; & \mathrm{Holder}’s\mathrm{inequality}\mathrm{for}\\ \; & \; & t=\alpha ,s=\frac{\alpha }{\alpha -1} \end{array}$$
$$\begin{array}{ccc} \;\;\;\;\;\;\;\;\;\;\;\;\;\;\;\;\;\;\;\;\;\; & ={\left(\sum_{x\in X}\frac{q{\left(x\right)}^{\alpha }}{p{\left(x\right)}^{\alpha -1}}\right)}^{\frac{1}{\alpha }}{\left(\sum_{x\in X}p\left(x\right)L{\left(h,f\right)}^{\frac{\alpha }{\alpha -1}}\right)}^{\frac{\alpha -1}{\alpha }}\; & \\ \; & ={\left({\left(\sum_{x\in X}\frac{q{\left(x\right)}^{\alpha }}{p{\left(x\right)}^{\alpha -1}}\right)}^{\frac{1}{\alpha -1}}\right)}^{\frac{\alpha -1}{\alpha }}{\left(\sum_{x\in X}p\left(x\right)L{\left(h,f\right)}^{\frac{\alpha }{\alpha -1}}\right)}^{\frac{\alpha -1}{\alpha }}\; & \\ \; & ={\left(d_{\alpha }\left(q||p\right)\right)}^{\frac{\alpha -1}{\alpha }}{\left(\sum_{x\in X}p\left(x\right)L{\left(h,f\right)}^{\frac{\alpha }{\alpha -1}}\right)}^{\frac{\alpha -1}{\alpha }}\; & \mathrm{By}\mathrm{Definition}\;3 \end{array}$$For each i∈{1,...,k}, by setting $p:=p_{i},q:=\hat{p_{i}}$ and $h:=h_{i}$, we will find that:
$$L\left(h_{i},\hat{p_{i}},f\right)\geq {\left(d_{\alpha }(\hat{p_{i}}\left|\right|p_{i})\right)}^{\frac{\alpha -1}{\alpha }}{\left(\sum_{x\in X}p_{i}\left(x\right)L{\left(h_{i},f\right)}^{\frac{\alpha }{\alpha -1}}\right)}^{\frac{\alpha -1}{\alpha }}={\left(d_{\alpha }(\hat{p_{i}}\left|\right|p_{i})\right)}^{\frac{\alpha -1}{\alpha }}\psi$$□

We contend that the value of $\psi ={\left(\sum_{x\in X}p_{i}\left(x\right)L{\left(h_{i},f\right)}^{\frac{\alpha }{\alpha -1}}\right)}^{\frac{\alpha -1}{\alpha }}$ can be disregarded when examining the loss bound. As previously mentioned, we assume that $L(h_{i},f)\leq M$, where we have set M=1. Consequently, we are left with ${\left(\sum_{x\in X}p_{i}\left(x\right)\right)}^{\frac{\alpha -1}{\alpha }}$. Since $p_{i}$ is a distribution, the sum equals 1.

Let us set $L_{\alpha }\left(\hat{p},p\right):={\left(d_{\alpha }(\hat{p}\left||p\right)\right)}^{\frac{\alpha -1}{\alpha }}$. We would like to present an example of different $L_{\alpha }\left(\hat{p},p\right)$ values calculated with a constant distribution p∼N(3,10), and a distribution $\hat{p}∼N\left(\mu ,10\right)$, where 0<μ<6. When μ=3, $p=\hat{p}$. The results are shown in Figure 5.

As we can observe, as the estimated distribution $\hat{p}$ approaches the true distribution *p* (i.e., as μ approaches 3), the bounds on the loss function become increasingly similar. We can also see that the value of the lower bounds is not significantly large, which means that we can consider using the probability approximation to solve the MSA problem. It is also worth noting that when α deviates significantly from 1, the bounds move away from the actual value.

**Theorem** **9.**
*Let $p_{T}$ be an arbitrary target distribution. For any δ>0, there exists η>0 and w∈Δ, such that the following inequality holds for any α>1 and any mixture parameter λ:*

(46)
$$L\left(h_{w}^{\eta },p_{T},f\right)\leq {\left(\left(\epsilon +\delta \right)d_{\alpha }(p_{T}\left|\right|p_{\lambda })\right)}^{\frac{\alpha -1}{\alpha }}M^{\frac{1}{\alpha }}$$



**Proof of Theorem** **9.**Let δ>0. In the proof for Corollary 1, we showed that for any hypothesis *h* and any distributions *p*, *q*, and for any α>1, the following holds:
(47)$$L\left(h,q,f\right)\leq {\left(d_{\alpha }\left(q||p\right)L\left(h,p,f\right)\right)}^{\frac{\alpha -1}{\alpha }}M^{\frac{1}{\alpha }}$$ Hence, for $q=p_{T}$, $p=p_{\lambda }$ and $h=h_{w}^{\eta }$ we will find that:
(48)$$L\left(h_{w}^{\eta },p_{T},f\right)\leq {\left(d_{\alpha }(p_{T}\left|\right|p_{\lambda })L\left(h_{w}^{\eta },p_{\lambda },f\right)\right)}^{\frac{\alpha -1}{\alpha }}M^{\frac{1}{\alpha }}$$ By Theorem 6, given δ>0, there exist η>0 and w∈Δ such that $L(h_{w}^{\eta },p_{\lambda },f)\leq \epsilon +\delta$ for any mixture parameter λ. Therefore:
(49)$$L\left(h_{w}^{\eta },p_{T},f\right)\leq {\left(d_{\alpha }(p_{T}\left|\right|p_{\lambda })\left(\epsilon +\delta \right)\right)}^{\frac{\alpha -1}{\alpha }}M^{\frac{1}{\alpha }}$$□

**Corollary** **3.**
*Let $p_{T}$ be an arbitrary target distribution. For any δ>0, there exists η>0 and w∈Δ, such that the following inequality holds for any α>1 and any mixture parameter λ∈Δ:*

(50)
$$L\left({\hat{h}}_{w}^{\eta },p_{T},f\right)\leq {\left(\left(\epsilon^{*}+\delta \right)d_{\alpha }(p_{T}\left|\right|{\hat{p}}_{\lambda })\right)}^{\frac{\alpha -1}{\alpha }}M^{\frac{1}{\alpha }}$$

*where ${\hat{p}}_{\lambda }=\sum_{i=1}^{k}\lambda_{i}{\hat{p}}_{i}\left(x\right)$ and ${\hat{h}}_{w}^{\eta }$ is our good hypothesis from Definition 9 but calculated with the estimated probabilities ${\hat{p}}_{i}$.*


**Proof of Corollary** **3.**By Corollary 1, ∀i∈{1,...,k} and for any α>1: $L\left(h_{i},\hat{p_{i}},f\right)\leq {\left(d_{\alpha }(\hat{p_{i}}\left|\right|p_{i})\epsilon \right)}^{\frac{\alpha -1}{\alpha }}M^{\frac{1}{\alpha }}$. Let us set $\epsilon^{*}$ such that: $\epsilon^{*}=\max_{i=1}^{k}\left\{{\left(d_{\alpha }(\hat{p_{i}}\left|\right|p_{i})\epsilon \right)}^{\frac{\alpha -1}{\alpha }}M^{\frac{1}{\alpha }}\right\}$. Overall, we obtained the following:For every i∈{1,...,k}: $L\left(h_{i},\hat{p_{i}},f\right)\leq \epsilon^{*}$.${\hat{h}}_{w}^{\eta }\left(x,y\right)=\sum_{i=1}^{k}\frac{w_{i}{\hat{p}}_{i}\left(x\right)+\eta \frac{U(x)}{k}}{\sum_{j=1}^{k}\left(w_{j}{\hat{p}}_{j}\left(x\right)\right)+\eta U\left(x\right)}h_{i}\left(x,y\right)$. We can repeat the proof of Theorem 9 with $\epsilon^{*}$ instead of ϵ, ${\hat{p}}_{i}$ instead of $p_{i}$ and ${\hat{h}}_{w}^{\eta }$ instead of $h_{w}^{\eta }$. □

In summary, we demonstrated that it is possible to use approximate distributions to calculate a good distribution-weighted combining rule. We have established that the error introduced by using estimated distributions is bounded. Thus, we can address the Multi-Source Adaptation (MSA) problem in real-world applications.

### 4.4. MSA Algorithm

Alongside the unknown probabilities, another crucial aspect is determining an appropriate vector of weights, denoted as *w*, to fully establish the distribution-weighted combining rule. The paper by Hoffman, Mohri, and Zhang [9] presents a new algorithm for determining the distribution-weighted combination solution for cross-entropy loss and other losses, based on Difference of Convex (DC) programming.

**Lemma** **2.**
*For any target function f∈F and any $\eta ,\eta^{\prime }\geq 0$, there exists w∈Δ with $w_{i}\neq 0$ for all i∈{1,...,k}, such that the following holds:*

(51)
$$\forall i\in \left\{1,...,k\right\}\;\;L\left(h_{w}^{\eta },p_{i},f\right)\leq \gamma +\eta^{\prime }$$

*where:$\gamma =\sum_{j=1}^{k}w_{j}L\left(h_{w}^{\eta },p_{j},f\right)$*


The proof of Lemma 2 is detailed in [19].

**Corollary** **4.**
*For any target function f∈F and any $\eta^{\prime }\geq 0$, there exists w∈Δ with $w_{i}\neq 0$ for all i∈{1,...,k}, such that the following holds:*

(52)
$$L\left(h_{w}^{\eta },p_{i},f\right)\leq L\left(h_{w}^{\eta },p_{w},f\right)+\eta^{\prime }\;\;\;\;\forall i\in \left\{1,...,k\right\}$$



**Proof of Corollary** **4.**By Lemma 2, we obtain:
$$\begin{array}{cc} \forall i\in \left\{1,...,k\right\}\;\;L\left(h_{w}^{\eta },p_{i},f\right) & \leq \gamma +\eta^{\prime }\\ \; & =\sum_{j=1}^{k}w_{j}L\left(h_{w}^{\eta },p_{j},f\right)+\eta^{\prime }\\ \; & =L\left(h_{w}^{\eta },p_{w},f\right)+\eta^{\prime } \end{array}$$□

Corollary 4 provides a single upper bound for the loss with respect to every $p_{i}$. Thus, our problem consists of finding a parameter *w* verifying this property. This, in turn, can be formulated as the following optimization problem:(53)$$\min_{w\in \Delta ,\rho \in R}\;\;\rho \;\;s.t.\;\;L\left(h_{w}^{\eta },p_{i},f\right)-L\left(h_{w}^{\eta },p_{w},f\right)\leq \rho \;\;\forall i\in \left\{1,...,k\right\}$$

**Definition** **10.**
*DC Function [23]: Let C be a convex subset of $R^{n}$. A real-valued function f:C→R is called DC on C, if there exist two convex functions g,h:C→R such that f can be expressed in the form:*

(54)
f(x)=g(x)−h(x)



DC programming problems are programming problems dealing with DC functions. An important class of DC problems is the following:(55)$$w^{*}=\inf\left\{g\left(x\right)-h\left(x\right)\;\;:x\in X\right\}$$
where *g* and *h* are two convex functions in $R^{n}$, and *X* is a closed convex subset of $R^{n}$.

**Proposition** **3.**
*Assume that the problem $w^{*}$ is solvable. Then, a point $x^{*}\in X$ is an optimal solution to $w^{*}$ if and only if there is $t^{*}\in R$, such that:*

(56)
$$0=\inf\{-h\left(x\right)+t\;\;:x\in X\;\;,t\in R\;\;,g\left(x\right)-t\leq g\left(x^{*}\right)-t^{*}\}$$



Horst and Thoai [23] developed an algorithm for solving DC programming problems such as $w^{*}$ based on the above optimality condition. The assumptions in Proposition 3 apply to the MSA problem, since we know there is an optimal solution. The key lies in identifying two convex functions whose difference coincides with the solution of the MSA problem. Let us define the following functions:(57)$$\begin{array}{cc} J_{w}\left(x,y\right) & =\sum_{i=1}^{k}w_{i}p_{i}\left(x\right)h_{i}\left(x,y\right)+\frac{\eta }{k}U\left(x\right)h_{i}\left(x,y\right)\\ K_{w}\left(x\right) & =p_{w}\left(x\right)+\eta U\left(x\right) \end{array}$$ Note that: $h_{w}^{\eta }\left(x,y\right)=\frac{J_{w}\left(x,y\right)}{K_{w}\left(x\right)}$.

Let us define the following convex functions: (58)$$\begin{array}{cc} u_{i}\left(w,f\right) & =-\sum_{x}\left[p_{i}\left(x\right)+\eta U\left(x\right)\right]\log\left(J_{w}\left(x,f\left(x\right)\right)\right) \end{array}$$(59)$$\begin{array}{cc} v_{i}\left(w,f\right) & =\sum_{x}K_{w}\left(x\right)\log\left(\frac{K_{w}\left(x\right)}{J_{w}\left(x,f\left(x\right)\right)}\right)-\left[p_{i}\left(x\right)+\eta U\left(x\right)\right]\log\left(K_{w}\left(x\right)\right) \end{array}$$

$u_{i}\left(w,f\right)$ is convex since $-\log(J_{w})$ is convex as a composition of the convex function −log with an affine function $J_{w}$. Similarly, $-\log(K_{w})$ is convex, which shows that the second term in the expression of $v_{i}\left(w,f\right)$ is a convex function. The first term can be written in terms of the unnormalized relative entropy (the unnormalized relative entropy of P and Q is defined by: $B\left(p||q\right)=\sum_{x}p\left(x\right)\log\left(\frac{p(x)}{q(x)}\right)+\sum_{x}\left(q(x)-p(x)\right)$). It can be shown that the relative entropy is jointly convex using the so-called log-sum inequality (based on the explanation in [9]).

Let us be reminded of our regression loss function:(60)$$\begin{array}{cc} L(h,p,f) & :=E_{x∼p}\left[L\left(h,f\right)\right]=\sum_{x\in X}L\left(h,f\right)p\left(x\right) \end{array}$$(61)$$\begin{array}{cc} L(h,f) & :=-\log h(x,f(x)) \end{array}$$

**Proposition** **4.**
*Let L be the cross-entropy loss. Then, for i∈{1,...,k}*

(62)
$$L\left(h_{w}^{\eta },p_{i},f\right)-L\left(h_{w}^{\eta },p_{w},f\right)=u_{i}\left(w,f\right)-v_{i}\left(w,f\right)$$



**Proof of Proposition** **4**

$$\begin{array}{ccc} \; & L\left(h_{w}^{\eta },p_{i},f\right)-L\left(h_{w}^{\eta },p_{w},f\right)\\ \; & =\sum_{x\in X}L\left(h_{w}^{\eta },f\right)p_{i}\left(x\right)-\sum_{x\in X}L\left(h_{w}^{\eta },f\right)p_{w}\left(x\right)\\ \; & =\sum_{x\in X}\left(p_{i}\left(x\right)-p_{w}\left(x\right)\right)L\left(h_{w}^{\eta },f\right)\\ \; & =\sum_{x\in X}\left(p_{i}\left(x\right)-p_{w}\left(x\right)\right)\left(-\log(h_{w}^{\eta }\left(x,f\left(x\right)\right))\right) & L\mathrm{is}\mathrm{the}\mathrm{cross}\mathrm{entropy}\mathrm{loss}.\\ \; & =\sum_{x\in X}\left(p_{i}\left(x\right)-p_{w}\left(x\right)\right)\left(-\log(\frac{J_{w}\left(x,f\left(x\right)\right)}{K_{w}\left(x\right)})\right) & h_{w}^{\eta }\left(x,y\right)=\frac{J_{w}\left(x,y\right)}{K_{w}\left(x\right)}\\ \; & =\sum_{x\in X}\left(p_{i}\left(x\right)-K_{w}\left(x\right)+\eta U\left(x\right)\right)\left(-\log(\frac{J_{w}\left(x,f\left(x\right)\right)}{K_{w}\left(x\right)})\right) & K_{w}\left(x\right)=p_{w}\left(x\right)+\eta U\left(x\right)\\ \; & =\sum_{x\in X}K_{w}\left(x\right)\left(\log(\frac{J_{w}\left(x,f\left(x\right)\right)}{K_{w}\left(x\right)})\right)\\ \; & \;\;\;\;\;\;-\sum_{x\in X}\left(p_{i}\left(x\right)+\eta U\left(x\right)\right)\left(\log(\frac{J_{w}\left(x,f\left(x\right)\right)}{K_{w}\left(x\right)})\right)\\ \; & =-\sum_{x\in X}K_{w}\left(x\right)\left(\log(\frac{K_{w}\left(x\right)}{J_{w}\left(x,f\left(x\right)\right)})\right)\\ \; & \;\;\;\;\;\;-\sum_{x\in X}\left(p_{i}\left(x\right)+\eta U\left(x\right)\right)\left(\log(J_{w}\left(x,f\left(x\right)\right)\right)\\ \; & \;\;\;\;\;\;+\sum_{x\in X}\left(p_{i}\left(x\right)+\eta U\left(x\right)\right)\left(\log(K_{w}\left(x\right))\right)\\ \; & =u_{i}\left(w,f\right)-v_{i}\left(w,f\right) & \mathrm{By}\mathrm{Equation}\;(58) \end{array}$$

□

Using the proof above, our optimization problem
$$\min_{w\in \Delta ,\rho \in R}\;\;\rho \;\;s.t.\;\;L\left(h_{w}^{\eta },p_{i},f\right)-L\left(h_{w}^{\eta },p_{w},f\right)\leq \rho \;\;\forall i\in \left\{1,...,k\right\}$$
is a DC programming problem, since it is the difference between two convex functions. In light of all of the above, our optimization problem can be cast as the following variational form of a DC-programming problem: let us set $\left(w_{t}\right)$ to be the sequence defined by repeatedly solving the following convex optimization problem:Target function: minρ.Constraints:$u_{i}\left(w,f\right)-v_{i}\left(w,f\right)-\left(w-w_{t}\right)\nabla v_{i}\left(w_{t},f\right)\leq \rho$$\sum_{i=1}^{k}w_{i}-1=0$$-w_{i}\leq 0\;\;\;\forall i\in \left\{1,...,k\right\}$ where $w_{0}\in \Delta$ is an arbitrary starting value. Then, $\left(w_{t}\right)$ is guaranteed to converge to a *local minimum* of the optimization problem [9].

Given the fact that an optimal hypothesis $h_{w}^{\eta }$ exists, we converted the MSA problem into an optimization problem and cast it to a DC programming form in order to find a local optimum. This way, we are able to find the parameter *w* which is used in the distribution-weighted combination rule.

## 5. Empirical Results

In this section we present two sets of experiments. The first set is designed to evaluate the accuracy of approximating distributions using the VRLU and VRS methods, and the second set demonstrates the application of these estimates for the MSA problem.

### 5.1. VRLU and VRS Experiments

We present a series of experiments conducted to evaluate the performance of VRLUα and $\mathrm{VRS}\alpha_{+},\alpha_{-}$ and compare them to the performance of existing methods such as the Evidence Lower Bound (ELBO) and Rényi upper and lower bounds. The goal of these experiments is to assess the effectiveness of the proposed methods and to determine their advantages and limitations. The methods we will examine in this section are detailed below:**VAE**—minimizing KL divergence—maximizing the ELBO.**VR**—minimizing Rényi divergence using variational Rényi upper / lower bound with MC, for different values of α.**VRLU**—minimizing Rényi divergence using our variational Rényi log upper bound with MC for different values of negative α.**VRS**—minimizing Rényi divergence using the (sandwich) upper-lower bound with MC for different values of negative and positive α. All of our experiments were conducted using PyTorch. Throughout the experiments, we used K = 50 samples for Monte Carlo (MC) approximation; trained the VAE models using the ADAM optimizer [24]; and set the learning rate to 0.001 and the batch size to 128 for the training set, and 32 for the test set. Our VAE model includes a total of 6 linear layers. The first 3 are the encoder layers, and the last 3 are the decoder layers. The dimension of the latent space is 50. We suggest two perspectives to evaluate and compare performances:*Quality of the decoded signal*—Reconstruction error, measured by Mean Square Error (MSE) and Cross-Entropy (CE).*Quality of the evidence approximation*—Maximizing the evidence log-likelihood, logp(x); the higher the better.

#### 5.1.1. Digits Experiment

In the following experiment, we used the ‘MNIST’, ‘USPS’, and ‘SVHN’ datasets, all of which contain digit images (See Figure 6). They all share 10 classes of digits. The ‘USPS’ dataset consists of 7291 training images and 2007 test images of size 16×16. The ‘MNIST’ dataset consists of 60,000 training images and 10,000 test images of size 28×28. ‘SVHN’ is obtained from house numbers in Google Street View images. It has 73,257 training images and 26,032 test images of size 32×32. If we look at Figure 6, we can see that the graphical representation of digits in ‘USPS’, ‘SVHN’, and ‘MNIST’ is very diverse; hence, each domain has a very different distribution.

We compared the learning curves of ${\mathrm{VRS}}_{\alpha_{+},\alpha_{-}}$ with $\alpha_{-}\in \left\{-0.5,-2\right\}$ and $\alpha_{+}\in \left\{0.5,2\right\}$ and ${\mathrm{VR}}_{\alpha }$ with α∈{0.5,2,5} over the ‘MNIST’ dataset. Figure 7 demonstrates that ${\mathrm{VRS}}_{\alpha_{+},\alpha_{-}}$ converged faster than ${\mathrm{VR}}_{\alpha }$ and the resulting loss value is smaller for both α values. Also, we can see that ${\mathrm{VR}}_{0.5}$ performs better than ${\mathrm{VR}}_{2}$, and ${\mathrm{VR}}_{2}$ performs better than ${\mathrm{VR}}_{5}$. This observation is in sync with the results reported in [3].

Figure 8 depicts the mean squared error (MSE) for the different learning methods. We can see that the MSE reconstruction error of all Variational Rényi methods, and specifically ${\mathrm{VRS}}_{0.5,-0.5}$, are better than VAE reconstruction error in all of the datasets.

#### 5.1.2. Faces Experiment

We performed a similar experiment on a dataset of facial expressions known as PIE. The PIE dataset consists of a few parts, each corresponding to a different posture. Specifically, we choose PIE05 (left pose), PIE07 (top pose), and PIE09 (bottom pose). In each subset (pose), all face images were taken under different lighting, illumination, and expression conditions (see Figure 9).

We divided each dataset into training and testing sets, in a ratio of 2:1. We created VAE, ${\mathrm{VR}}_{\alpha }$ and ${\mathrm{VRS}}_{\alpha_{+},\alpha_{-}}$ models for each ‘PIE’ domain (left pose, up position, and down position). Each model was trained on its corresponding training set. We calculated the log-likelihood estimations for each domain and compared them. The results are presented in Figure 10. We can see that the ${\mathrm{VRS}}_{\alpha_{+},\alpha_{-}}$ model achieved the best results. In addition, for $\alpha_{+}=0.5$, we obtained slightly better results than for $\alpha_{+}=2$, which is compatible with all previous results.

To summarize, we demonstrated the performance of the ${\mathrm{VRS}}_{\alpha_{+},\alpha_{-}}$ algorithm on the digits datasets (‘MNIST’, ‘USPS’, ‘SVHN’) and ‘PIE’ datasets (left pose, up position, and down position), and compared them against the (KL divergence-based) VAE, the Variational Rényi ${\mathrm{VR}}_{\alpha }$ upper and lower bounds, and the ${\mathrm{VRLU}}_{\alpha }$ upper bound minimization. In all cases, the ${\mathrm{VRS}}_{\alpha_{+},\alpha_{-}}$ algorithm presented good results, many of which are the best performances compared to the other methods.

### 5.2. MSA Experiments

In this section, we review a series of experiments designed to tackle the MSA problem. In all of the experiments, we used the DC-programming algorithm, presented in [9], to provide a solution. We used real-world datasets: the digit dataset and Office31 dataset. For all of the datasets, the probability distributions $p_{i}$ are not readily available to the learner. Thus, we used the VAE, ${\mathrm{VR}}_{\alpha }$ and ${\mathrm{VRS}}_{\alpha_{+},\alpha_{-}}$ models to approximate the probabilities ${\hat{p}}_{i}$. More concretely, given an MSA scenario, where we have *k* source domains and one target domain, we train a variational inference model for each source domain *i*. We then use the estimated distributions as input to the DC programming algorithm, which, in turn, finds the optimal vector *w* used to construct the distribution-weighted combination hypothesis $h_{w}^{\eta }$ (Definition 9) for the target domain. We term the technique described above as **VRS-MSA**. Finally, we compared our performances to the results presented by Cortes et al. in [20].

#### 5.2.1. Digits Experiment

In the following experiment, we used the digits datasets, SVHN, MNIST, and USPS, as our source domains. For each domain, we trained a convolutional neural network (CNN) of the same architecture as in [25], and used the output from the softmax score layer as our base predictors $h_{i}$. We also trained the VAE, ${\mathrm{VR}}_{\alpha }$ and ${\mathrm{VRS}}_{\alpha_{+},\alpha_{-}}$ models for each domain using the respective training sets. We used these trained models to approximate the domains’ distributions ${\hat{p}}_{i}$.

For the DC-programming algorithm, we used 1000 image–label pairs from each domain, thus being a total of 3000 labeled pairs, to learn the parameter *w*. We compared our **VRS-MSA** algorithm against the results presented in [20], and report performances on each of the three test datasets, on combinations of two test datasets, and on all test datasets combined.

Table 2 details the accuracy scores obtained by running our **VRS-MSA** model and the following models:CNN-s, CNN-m, and CNN-u: each trained on the single source domain SVHN, MNIST, and USPS, respectively.CNN-unif: a classifier trained on a uniform combination of the source domains’ data.CNN-joint: a global classifier trained on all of the source domains’ data combined.The GMSA model: a generative MSA model using the DC programming algorithm. To obtain the data distribution, GMSA used the last layer before softmax from each of the domains’ classifiers.The DMSA model: this is based on a discriminative technique using an estimate of the conditional probabilities (the probability that point *x* belongs to source *i*).

Our **VRS-MSA** model demonstrates competitive performance, with particularly strong results on the union of the SVHN and MNIST test sets and the union of the SVHN, MNIST, and USPS test sets. Moreover, among VI models, ${\mathrm{VRS}}_{0.5,-0.5}$ achieved the best average score. This result is consistent with our previous results, which state that the closer α is to zero, the better the approximation of the log evidence.

However, the performance on the SVHN domain is lower in comparison to the other classifiers. Taking a closer look at the parameter $w=(w_{{}_{MNIST}}:0.73,w_{{}_{USPS}}:0.19,w_{{}_{SVHN}}:0.08)$ reveals that the value assigned to the SVHN domain, denoted as $w{}_{SVHN}$, is relatively low at 0.08. Since the distribution weighted combining rule is a weighted combination of all source hypotheses with weights assignment *w*, this indicates that the SVHN domain has a minimal impact on the calculation of $h_{w}^{\eta }$. Additionally, the log probability obtained for the SVHN domain using the VI models is quite low compared to the other domains. These low values result in very small probabilities when taking the exponent, which can be difficult to work with in practice.

Furthermore, we devised a method that uses Stochastic Gradient Descent (SGD), rather than DC programming, to get a good classifier for the target domain. For each image *x*, every possible label $y_{1},...,y_{c}$, and every source domain, we created the following input:$$\left(p_{1}\left(x,y_{1}\right),...,p_{1}\left(x,y_{c}\right),...,p_{k}\left(x,y_{1}\right),...,p_{k}\left(x,y_{c}\right),h_{1}\left(x,y_{1}\right),...,h_{1}\left(x,y_{c}\right),...,h_{k}\left(x,y_{1}\right),...,h_{k}\left(x,y_{c}\right)\right)$$ Given image x, the SGD model learns a matching between the input vector above and the true label of x. This method is termed **VRS-SGD**. Similarly to VRS-MSA, we used 1000 images from each domain to train the SGD model. The results of the VRS-SGD are reported at the last section of Table 2.

The SGD score for the SVHN test set stands out as the highest, leading to an improvement in the combined test set that includes both SVHN and USPS. One advantage of the VRS-SGD method is its ability to overcome the issue of misalignment among different VRS models by adjusting its learned weights to match the input scale. This makes the VRS-SGD method particularly valuable when working with source domains where the probabilities are smaller compared to other domains.

#### 5.2.2. Office Experiment

In the following experiment, we used the Office31 dataset, which is used mainly in domain adaptation scenarios. The Office31 dataset contains 31 object categories in three domains: Amazon, DSLR, and Webcam (see Figure 11). The 31 categories in the dataset consist of objects commonly encountered in office settings, such as keyboards, file cabinets, and laptops. The Amazon domain contains on average 90 images per class and 2817 images in total. As these images were captured from a website of online merchants, they are captured against a clean background and at a unified scale. The DSLR domain contains 498 low-noise high-resolution images (4288 × 2848). There are 5 objects per category. Each object was captured from different viewpoints on average 3 times. For Webcam, the 795 images of low resolution (640 × 480) exhibit significant noise and color as well as white balance artifacts.

We carried out the **VRS-MSA** experiment on Office31 dataset. We divided the dataset into two splits following [26]. For the training data, we used 20 samples per category for Amazon and 7 for both DSLR and Webcam. We used the rest of the samples as test data. For each domain, we used ResNet50 architecture pre-trained on ImageNet, and trained it over the domain’s training set. We extracted the penultimate layer output from ResNet50 architecture and trained our variational inference models VAE, ${\mathrm{VR}}_{\alpha }$ and ${\mathrm{VRS}}_{\alpha_{+},\alpha_{-}}$ on this pre-trained feature. The VI models were used to approximate the distributions $p_{i}$. For our predictors $h_{i}$, we extracted the output from the ResNet50 architecture and used softmax layer to calculate the probabilities. We used a batch size of 32 in the training set and 16 in the test set.

We measured the performance of these baselines on each of the three test sets, on combinations of two test sets, and all test sets combined. We compared our **VRS-MSA** model against previous results presented by Cortes et al. [20]. While Cortes et al. only provided results for individual test sets, we additionally presented results for various combinations of test sets, providing a more comprehensive comparison of the performance of VI models. Among the models tested, our ${\mathrm{VRS}}_{0.5,-0.5}$ model achieved the highest results in most test set combinations and had the best overall score, which supports our previous findings that a value of α close to zero leads to a better approximation of the log evidence.

We compared our results to the DMSA algorithm, each source predictor (CNN for Amazon, DSLR and Webcam), the uniform combination, CNN-unif, a network jointly trained on all source data combined, CNN-joint, and GMSA with kernel density estimation [9]. The results are reported in Table 3.

Our **VRS-MSA** model demonstrates competitive achievements, with particularly strong results on the test set DSLR. We note that the DSLR’s high score comes at the expense of Amazon’s and Webcam’s high scores. This is because the vector $w=(w_{{}_{Amazon}}:0.25,w_{{}_{DSLR}}:0.71,w_{{}_{Webcam}}:0.04)$ learned in the DC programming algorithm determined the weight of each domain. When the DSLR domain receives more weight, it comes at the expense of the weight given to the other domains.

Likewise, the VRS-SGD method achieved competitive scores compared to the models using the DC algorithm. We can see that the VRS-SGD score for the Amazon test set is the highest and, as a result, the scores on test sets that include Amazon were also improved.

## 6. Summary

In this study, we reviewed and analyzed the methods to estimate data probabilities where traditional computation methods have failed. Specifically, we examined variational inference (VI) models, such as Variational Autoencoder (VAE) [27], which we aimed to improve using different divergence methods. We examined the properties of the Kullback–Leibler divergence, the Rényi divergence (which is essentially a family of divergences parameterized by α∈R), and the χ divergence. We derived the ELBO, the VR, and the CUBO bounds for the log evidence, and presented a new upper bound, termed VRLU, for which its MC approximation remains an upper. We used VRLU to devise a new (sandwiched) upper–lower bound variational inference method (VRS). The VRS loss function combines the VR lower bound (with positive α) and the new VRLU upper bound (with negative α), thus providing a tighter estimate for the log evidence.

We performed several experiments designed to test the performance of the new VRS model. We compared VAE, VR, VRLU, and VRS models over the digits datasets and PIE datasets, using different values of positive and negative α. In all cases, the VRS algorithm presented good results, many of which are the best performances compared to the other methods. We note, in passing, that the selection of the α value may depend on the data, an observation that was indicated in previous studies, as well [3,14].

In addition, we demonstrated the usage of VRS in MSA applications. We combined the DC-programming algorithm (suggested in [9]) with our VRS model, to obtain more accurate density estimates and improve the accuracy of the hypothesis for the target domain. We performed experiments to compare the accuracy of the resulting hypothesis in two MSA datasets: the digits and Office31 datasets. We compared our new model using VAE, VR, and VRS to the previous models, GMSA and DMSA, presented in [20].

Our empirical evaluation revealed that the proposed VRS-MSA model demonstrated competitive performance, and in certain instances even surpassed the performance of models reported in previous studies. Additionally, among the VI models tested, the VRS model achieved the highest overall score, which supports the conclusion that accurate probability estimates are necessary for the success of the weighted combination hypothesis $h_{w}^{\eta }$.

Nonetheless, it is important to note that the VRS-MSA model achieved lower scores in certain individual test sets, where the weight parameter *w* was assigned a low value for that particular domain. When the weight parameter is low, it is important to take into account both the probability $p_{i}\left(x\right)$ and the domain-specific hypothesis $h_{i}$. For example, if the image *x* is from the SVHN domain, the probabilities $p_{mnist}\left(x\right)$ and $p_{usps}\left(x\right)$ should be relatively low in comparison to $p_{svhn}\left(x\right)$, such that the value of $h_{svhn}$ is the most prominent in the weighted combination hypothesis. Our VRS-MSA model operates by training a VRS model for each domain, which learns its latent space vectors based on a Gaussian distribution, and outputs the probability in relation to these latent vectors $p_{\theta }\left(x|z\right)$. Consequently, for each domain, the Gaussian distribution may have slight variations in variance, which can influence the log evidence value output from the VRS model. Therefore, the DC programming model, which takes into account the probabilities from all domains simultaneously, may be affected by the different scales of the probability measurements across the domains.

Looking forward, further work is required to disentangle the complexities of the aforementioned VRS-MSA. Specifically, in this work, we have not formed a connection between the latent variables of each VRS model of the different domains. It will be interesting to see how such a connection (of normalization, scaling of the probability measurements, or latent space alignment) will affect the compatibility of the probabilities. In addition, some researchers suggest even using a common latent feature space in the autoencoder models [28]. Building such a network using our VRS loss might improve the results of the VRS-MSA model. However, it is worth noting that such a common model would lack the separation and privacy of domains that we have achieved using distinct VRS models.

We would also like to extend our experiments on the VRS model: First, it will be interesting to examine the different values of negative and positive α values and search for the best combination of $\alpha_{-}$ and $\alpha_{+}$. Second, since α may be data-dependent, it will be interesting to explore the possibility to make α a trainable parameter. It can also be used to adjust the degree of relative risk aversion. These directions are left for future research efforts.

## Figures and Tables

**Figure 1 entropy-25-01468-f001:**
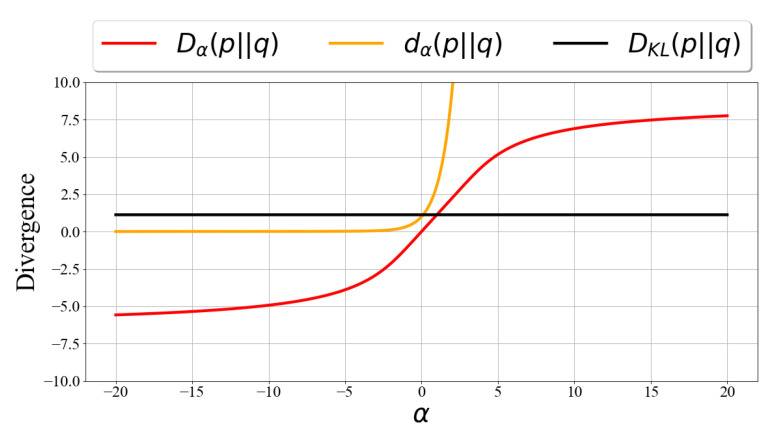
Illustration of $d_{\alpha }\left(p||q\right)$ vs. $D_{\alpha }\left(p||q\right)$ for fixed distributions p and q over different α values. p∼N(0,2), q∼N(3,2).

**Figure 2 entropy-25-01468-f002:**
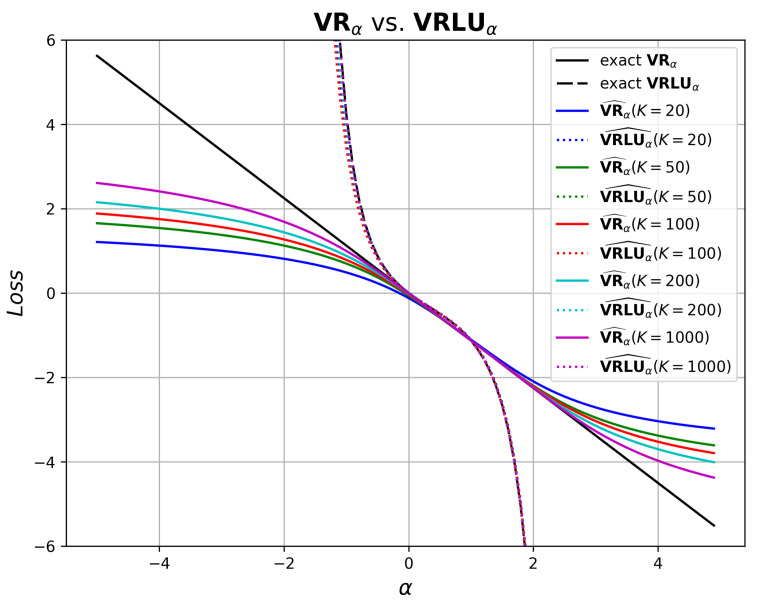
${\mathrm{VR}}_{\alpha }$ and ${\mathrm{VRLU}}_{\alpha }$, vs. their Monte Carlo approximations with different number of samples *K*, over a range of α values, using fixed distributions: p∼N(0,1) and q∼N(1.5,1).

**Figure 3 entropy-25-01468-f003:**
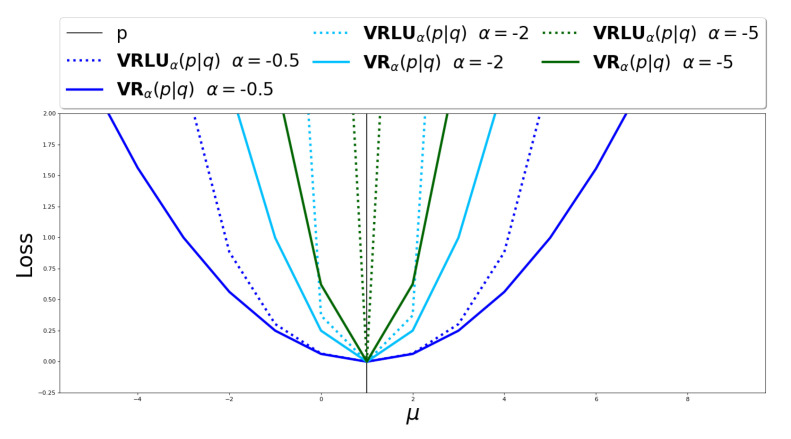
Comparison between ${\mathrm{VR}}_{\alpha }\left(p||q\right)$ and ${\mathrm{VRLU}}_{\alpha }\left(p||q\right)$ over different *q* distributions divergent from fixed distribution *p*. p=N(1,2) and q=N(μ,2) where −5≤μ≤10.

**Figure 4 entropy-25-01468-f004:**
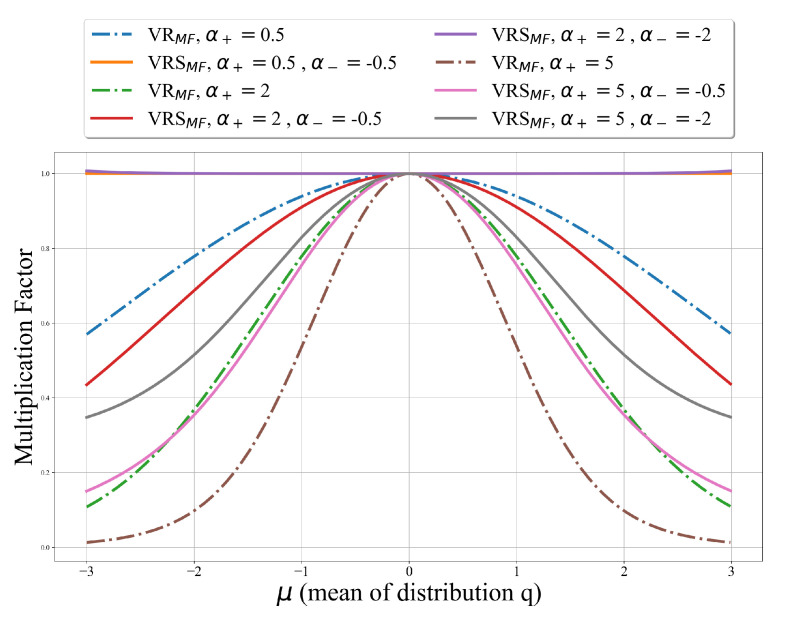
Comparison between ${\mathrm{VR}}_{\alpha }$ and ${\mathrm{VRS}}_{\alpha_{+},\alpha_{-}}$ multiplication factors over fixed distribution p and different q distributions.

**Figure 5 entropy-25-01468-f005:**
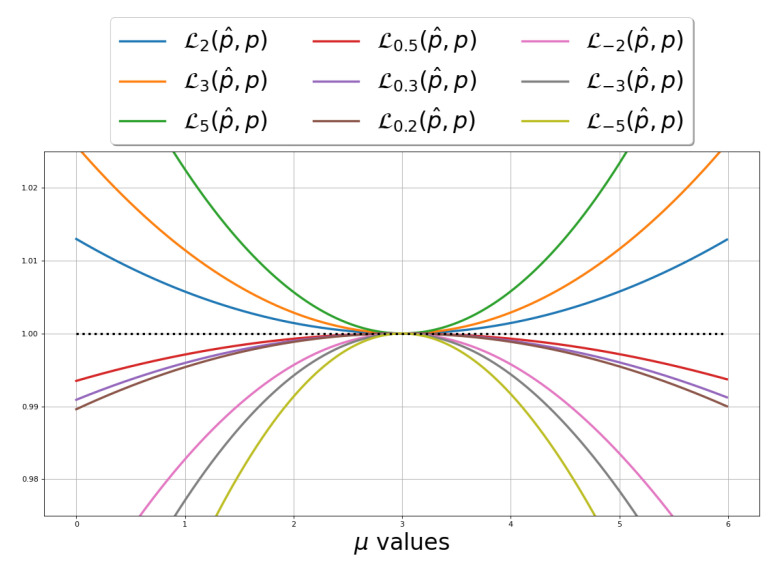
Comparison between $L_{\alpha }\left(\hat{p},p\right)$ with different α values over fixed distribution p∼N(3,10), and distribution $\hat{p}∼N\left(\mu ,10\right)$, where 0<μ<6.

**Figure 6 entropy-25-01468-f006:**
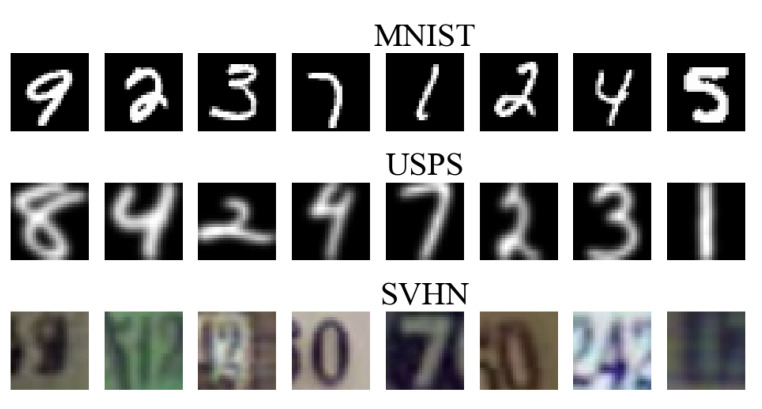
Digits datasets visualization.

**Figure 7 entropy-25-01468-f007:**
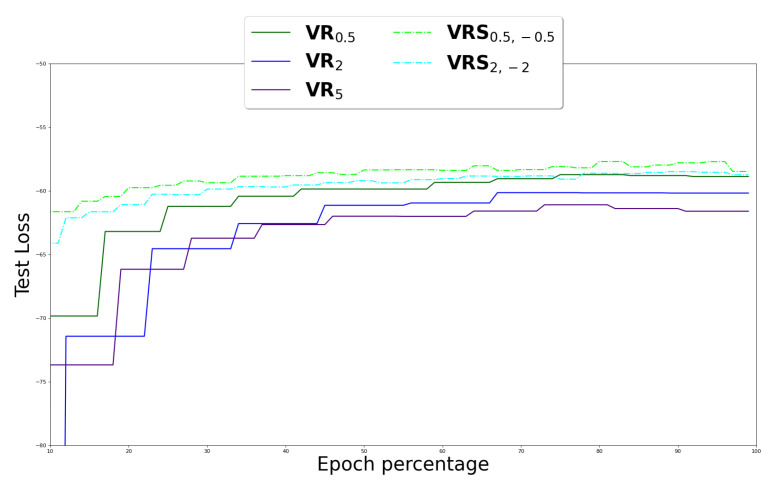
Comparison between ${\mathrm{VR}}_{\alpha }$ and ${\mathrm{VRS}}_{\alpha_{+},\alpha_{-}}$ learning curves over ‘MNIST’ dataset. Training with different values of α. The *y* axis detailed the values of the VR and VRS bounds, which is the approximation of the log evidence (the higher the better).

**Figure 8 entropy-25-01468-f008:**
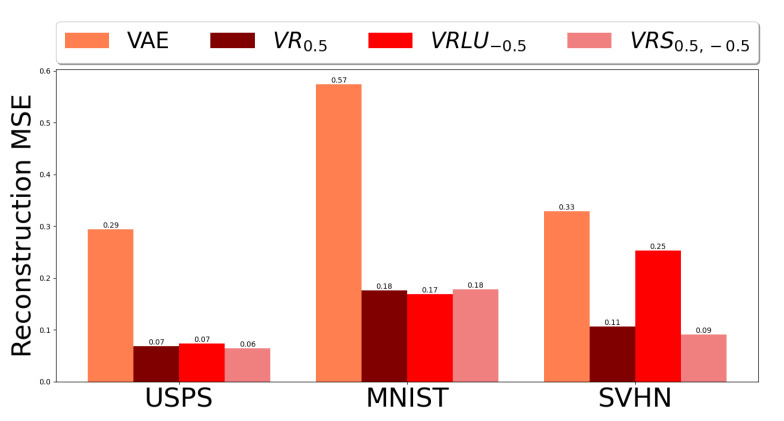
Comparison of the MSE values of VAE, ${\mathrm{VR}}_{0.5}$, ${\mathrm{VRLU}}_{-0.5}$ and ${\mathrm{VRS}}_{0.5,-0.5}$ over Digits datasets.

**Figure 9 entropy-25-01468-f009:**
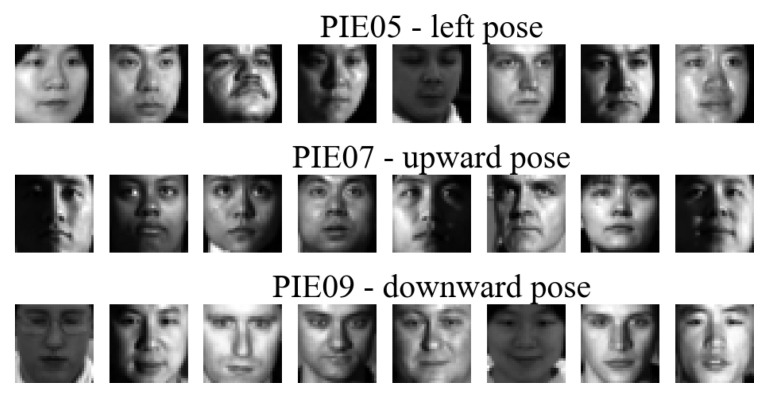
PIE datasets visualization.

**Figure 10 entropy-25-01468-f010:**
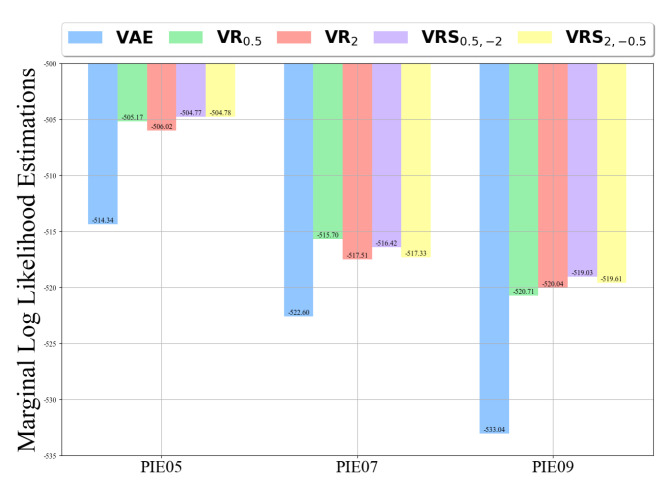
Comparison between the log-likelihood estimates, calculated using the models VAE, ${\mathrm{VR}}_{\alpha }$ and ${\mathrm{VRS}}_{\alpha_{+},\alpha_{-}}$ with different values of α. Each model was trained on a specific domain of ‘PIE’.

**Figure 11 entropy-25-01468-f011:**
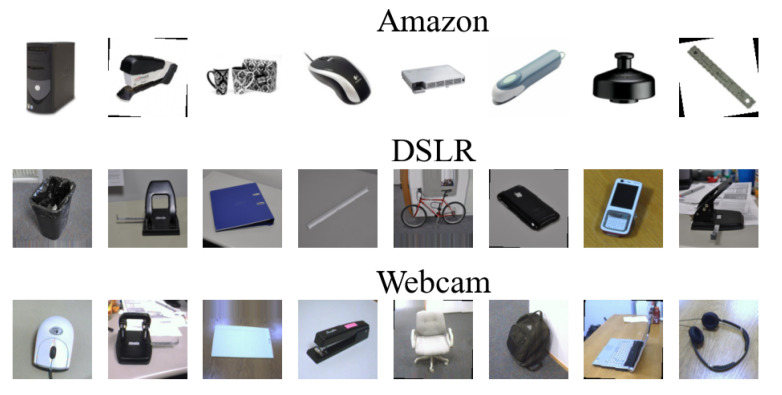
Office datasets visualization.

**Table 1 entropy-25-01468-t001:** Special cases in the Rényi divergence family.

α	*Definition *	*Notes*
α→0	−logq({p>0})	Not a divergence
α→1	$E_{x∼p(x)}\left[\log\frac{p(x)}{q(x)}\right]$	KL divergence
		Rényi divergence
$\alpha =\frac{1}{2}$	$-2\log\left(1-{\mathrm{Hel}}^{2}\left(p||q\right)\right)$	symmetric in
		its arguments
		Correlated to
α=2	$-\log\left(1-\chi^{2}\left(p||q\right)\right)$	the $\chi^{2}$
		divergence
α→∞	$\log\max\left(\frac{p}{q}\right)$	Worst-case regret

**Table 2 entropy-25-01468-t002:** Digit Dataset Accuracy (s—SVHN, m—MNIST and u—USPS). Previous results were taken from [20]. Bold labels signify the top score within the respective column.

Models	Test Datasets							
	**s**	**m**	**u**	**mu**	**su**	**sm**	**smu**	**Mean**
CNN-s	92.3	66.9	65.6	66.7	90.4	85.2	84.2	78.8
CNN-m	15.7	99.2	79.7	96.0	20.3	38.9	41.0	55.8
CNN-u	16.7	62.3	96.6	68.1	22.5	29.4	32.9	46.9
CNN-unif	75.7	91.3	92.2	91.4	76.9	80.0	80.7	84.0
CNN-joint	90.9	99.1	96.0	98.6	91.3	93.2	93.3	94.6
GMSA	91.4	98.8	95.6	98.3	91.7	93.5	93.6	94.7
DMSA	92.3	**99.2**	**96.6**	**98.8**	92.6	94.2	94.3	95.4
VAE-MSA	72.1	97.7	94.6	96.0	92.3	95.7	95.7	92.0
${VR}_{2}$-MSA	72.4	99.1	94.9	96.5	89.3	96.1	95.6	92.0
${VR}_{0.5}$-MSA	70.0	99.1	95.1	96.5	89.2	96.1	95.7	91.7
${VRS}_{2,-2}$-MSA	74.2	99.1	94.7	96.5	89.3	96.1	95.6	92.2
${VRS}_{2,-0.5}$-MSA	71.5	98.9	95.7	96.5	87.5	95.9	95.6	91.6
${VRS}_{0.5,-2}$-MSA	72.5	99.1	94.7	96.5	90.1	**96.1**	**95.7**	92.1
${VRS}_{0.5,-0.5}$-MSA	76.0	99.1	94.6	96.5	89.4	95.8	95.4	92.4
VAE-SGD	93.8	99.0	94.6	98.3	93.8	95.2	95.2	95.7
${VR}_{2}$-SGD	93.9	98.5	94.8	97.9	94.0	95.2	95.2	95.6
${VR}_{0.5}$-SGD	93.7	99.0	94.8	98.3	93.8	95.2	95.2	95.7
${VRS}_{2,-2}$-SGD	93.7	99.0	94.7	98.3	93.8	95.2	95.2	**95.7**
${VRS}_{2,-0.5}$-SGD	93.9	98.4	95.0	97.8	94.0	95.2	95.1	95.6
${VRS}_{0.5,-2}$-SGD	93.9	98.5	94.9	97.9	93.4	95.2	95.1	95.6
${VRS}_{0.5,-0.5}$-SGD	**93.9**	98.4	94.9	97.8	**94.0**	95.2	95.2	95.6

**Table 3 entropy-25-01468-t003:** Office Dataset Accuracy (a—Amazon, w—Webcam, d—DSLR). Previous results were taken from [20]. Bold labels signify the top score within the respective column.

Models	Test Datasets							
	**a**	**w**	**d**	**aw**	**ad**	**wd**	**awd**	**Mean**
Resnet-a	82.2	75.8	77.6	-	-	-	-	-
Resnet-w	63.3	95.7	95.7	-	-	-	-	-
Resnet-d	64.6	94.0	95.8	-	-	-	-	-
Resnet-unif	79.3	96.7	97.2	-	-	-	-	-
GMSA	82.1	96.8	96.7	-	-	-	-	-
DMSA	82.2	**97.2**	97.4	-	-	-	-	-
VAE-MSA	76.6	93.4	98.6	81.0	79.8	95.0	82.7	86.7
${VR}_{2}$-MSA	76.0	94.1	98.2	80.5	79.0	95.2	82.4	86.5
${VR}_{0.5}$-MSA	77.3	93.1	98.6	81.5	80.5	94.8	83.5	87.0
${VRS}_{0.5,-2}$-MSA	69.0	93.0	**99.0**	74.6	72.6	94.9	77.0	82.9
${VRS}_{2,-0.5}$-MSA	78.0	93.2	98.6	82.0	80.7	94.8	83.7	87.3
${VRS}_{2,-2}$-MSA	81.6	92.2	98.6	84.5	84.0	94.3	86.0	88.7
${VRS}_{0.5,-0.5}$-MSA	81.7	92.4	98.6	84.6	84.2	94.5	86.1	88.9
VAE-SGD	92.2	95.0	96.8	92.8	92.7	95.6	93.2	94.0
${VR}_{2}$-SGD	92.2	95.0	96.8	92.8	92.7	95.6	93.1	94.0
${VR}_{0.5}$-SGD	92.2	95.0	96.8	92.8	92.7	95.6	93.2	94.0
${VRS}_{2,-2}$-SGD	92.2	95.0	96.8	92.7	92.7	95.6	93.1	94.0
${VRS}_{2,-0.5}$-SGD	92.2	94.8	97.2	92.7	92.7	95.6	93.2	**94.1**
${VRS}_{0.5,-2}$-SGD	92.2	95.0	96.8	92.8	92.7	95.6	93.1	94.0
${VRS}_{0.5,-0.5}$-SGD	**92.2**	95.0	96.8	**92.8**	**92.7**	**95.6**	**93.2**	94.0

## Data Availability

https://github.com/DanaOshri/Multiple-Source-Adaptation-using-Variational-R-nyi-Bound-Optimization (accessed on 26 September 2023).

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
