# Peer review of "Variational Inference via Rényi Bound Optimization and Multiple-Source Adaptation†"

_entropy, 2023, doi:10.3390/e25101468_

Round 1

Reviewer 1 Report

"Variational Inference via Renyi Bound Optimization and Multiple-Source Adaptation" is an excellent article describing a novel method of training a VAE for use in adaptation to a non-trained target dataset. The authors provide a clear review of the use of Renyi and Chi-Square Divergence to modify the Evidence Lower-bound (ELBO) cost functions for VAE training.

From these methods they introduce a novel Variational Rényi Log Upper Bound (VRLU). They show that VRLU does a better job of assuring that that Monte Carlo approximates are similar to the exact solution. They then use both lower and upper bounds to define a Variational Renyi Sandwich (VRS) used to refine the control in training a VAE.

One area of the paper that requires improvement before publication is that they report their results only in terms of reconstruction loss. While reconstruction is the output of a VAE, its more general purpose is learning a probabilistic model. The accuracy of the model is measured by the ELBO, which includes both the reconstruction loss and the divergence between the latent posterior and prior distributions. Therefore, performance comparisons should also report the full ELBO and possibly the divergence.

Finally, there are a couple of comments in the paper that the proper value of alpha for VRS depends on the data. This is one use of alpha, to improve the accuracy of the model, but it is also possible to use alpha to improve the robustness which would depend on the relative risk regarding how outliers are handled. This second use case would be an additional constraint independent of the data.

The marked-up file expands upon these suggestions and includes a few minor editorial corrections. 

Author Response

Thank you for your detailed review. I learned a lot from your comments. I will update my paper and address your comments in my final version.

Only one question/clarification, when I described the second performance evaluation I said: Quality of the evidence approximation. This is exactly the ELBO for VAE, or the VR bound and the VRS bound. This is exactly the loss function, which I train the model based on. So, this evaluation takes in considuration both the reconstruction error and the divergence.

I do need to add some results for the MSE as well.

Thank you again for taking the time to review this manuscript,

 Dana.

Reviewer 2 Report

Major Comments:

1. Abstract Content: The abstract contains several subjective descriptions and judgments, such as "Their proof, however, is not constructive." These elements may not be appropriate for an abstract. Moreover, many introductions to other researchers' works are included in the abstract. These introductions would be more suitably placed in the main body's introduction section.

2. Repetitive Expressions: The first two sections of the manuscript seem to contain repeated expressions that are similar to existing works, specifically those cited as Ref 1,2,14. Furthermore, there is noticeable overlap with the authors' own conference proceedings. It is recommended to ensure originality and avoid using the exact same sentences from previous publications.

3. Lack of Theoretical Evidence: The paper heavily relies on empirical demonstrations without providing explicit proofs. Specifically, there is no clear explanation or justification regarding the "unbiased" nature of the VRLU. It would be beneficial to extend the discussion on this topic, especially when comparing it with the CUBO.

4. Dependence on Alpha: The VRS's applications in inferences appear to be largely dependent on alpha. This could potentially limit its applicability to different tasks. A more detailed discussion or guidance on the practical rules for choosing alpha would be valuable.

Minor Comment:

Figure 7 Clarification: There are some ambiguities in Figure 7 regarding the test loss. If it is indeed the MSE loss, as described by the authors, then one would expect the VRS to perform worse than the VR. This discrepancy should be clarified in the main text to prevent confusion.

It would be beneficial for the authors to address the above concerns to enhance the clarity, originality, and applicability of their work.

Author Response

Thank you for your detailed review. I will update the paper according to your comments.

Thanks,

Dana.

Round 2

Reviewer 1 Report

Yes, the revisions are acceptable.